# Personalizing Reinforcement Learning from Human Feedback with Variational Preference Learning

**Sriyash Poddar**[*], **Yanming Wan**[*], **Hamish Ivison, Abhishek Gupta**[†], **Natasha Jaques**[†]

Paul G. Allen School of Computer Science and Engineering
University of Washington
Seattle, WA 98195
`<sriyash, ymwan, hamishiv, abhgupta, nj>@cs.washington.edu`
`weirdlabuw.github.io/vpl`

## Abstract

Reinforcement Learning from Human Feedback (RLHF) is a powerful paradigm for aligning foundation models to human values and preferences. However, current RLHF techniques cannot account for the naturally occurring differences in individual human preferences across a diverse population. When these differences arise, traditional RLHF frameworks simply average over them, leading to inaccurate rewards and poor performance for individual subgroups. To address the need for pluralistic alignment, we develop a class of multimodal RLHF methods. Our proposed techniques are based on a latent variable formulation - inferring a novel user-specific latent and learning reward models and policies conditioned on this latent without additional user-specific data. While conceptually simple, we show that in practice, this reward modeling requires careful algorithmic considerations around model architecture and reward scaling. To empirically validate our proposed technique, we first show that it can provide a way to combat underspecification in simulated control problems, inferring and optimizing user-specific reward functions. Next, we conduct experiments on pluralistic language datasets representing diverse user preferences and demonstrate improved reward function accuracy. We additionally show the benefits of this probabilistic framework in terms of measuring uncertainty, and actively learning user preferences. This work enables learning from diverse populations of users with divergent preferences, an important challenge that naturally occurs in problems from robot learning to foundation model alignment.

## 1 Introduction

Reinforcement learning from human feedback (RLHF) has become the predominant technique for aligning AI foundation models to human values. Across domains like natural language processing (NLP) [52] to robotics [62, 50], RLHF is an effective way to improve the performance, accuracy, and safety of AI models, by ensuring that they align with human preferences [45, 34, 29] The question then becomes: whose preferences? Current RLHF approaches rely on a prescriptive set of values curated by a small set of AI researchers [39, 60, 2]. Moreover, they typically assume that all end-users share the same set of values. Given the concerning lack of diversity in AI [22], it is clear that this approach cannot account for the range of social, moral, and political values that inform preferences in populations. Recent work has demonstrated that with current RLHF techniques if the majority population has a weak preference for an outcome that will severely disadvantage a minority group,

---

[*][†] = Equal Contribution

38th Conference on Neural Information Processing Systems (NeurIPS 2024).

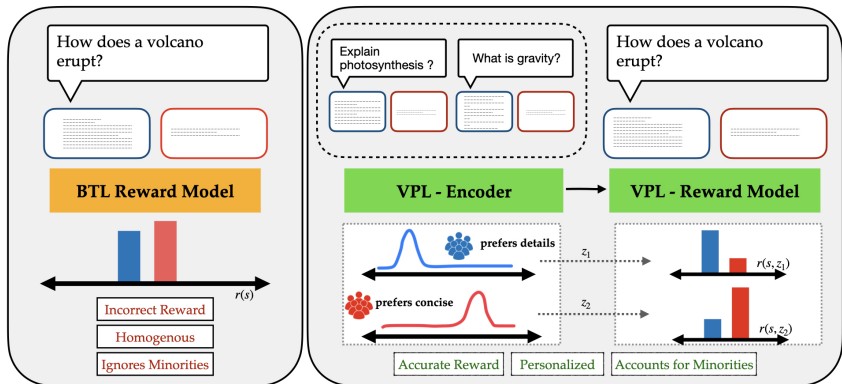

Figure 1: Current RLHF approaches [52] incorrectly assume a unimodal BTL reward model for a diverse population of users. In this example, users have diverging preferences over the level of detail provided in the responses from a large language model. Without additional context, the BTL model considers both responses to be equally likely. In contrast, our method, VPL, is a personalized approach to RLHF. Using a few samples from a particular user, we infer the distribution over their distinct preferences. Based on this distribution, we condition the reward model to more accurately predict rewards, and enable steering the resulting policy to personalize to the specific user. This enables accounting for and serving the preferences of under-represented groups which would otherwise be ignored by the standard BTL model [52].

the learned reward model will ignore the minority group's preferences [58]. These issues necessitate *pluralistic alignment* of models to human preferences [59, 44] (see Figure 1); ideally, we would democratize RLHF to account for a wider variety of human values to better serve a diverse population.

Current approaches to RLHF [52] use the Bradley-Terry-Luce (BTL) [15] model to learn a reward model that explains the human preferences. While the BTL model accounts for noisy preferences, RLHF typically applies this model under the 'unimodal' assumption that all human preferences are derived from a single utility function. This fails to capture scenarios where preferences diverge— i.e. are multi-modal—due to fundamentally different utilities. For example, Figure 1 shows a case where one group of users prefers detailed responses, while another prefers concise ones. By performing maximum likelihood estimation under the unimodal BTL model, current methods learn a reward function that averages these multi-modal preferences (akin to mode averaging in imitation learning [51]. As we show in our experimental results, this model misspecification leads to reward models that are *inaccurate*, and the policies optimized on these rewards fail to accomplish tasks per *any* of the distinct preferences (see Figures 6, 3). Thus, vanilla RLHF methods [52, 17] are insufficient for aligning AI systems to diverse human values.

In many applications, from large language models (LLMs) to assistive robotics [7], users are diverse, and AI systems must adapt the generated responses to user-specific preferences to successfully complete the task. Consider, for example, a robot assistant putting away dishes in a user's kitchen: each individual has unique preferences for how the dishes in their kitchen are organized, potentially diverging from others' preferences. In the context of LLMs, failing to adapt to user-specific preferences can make them unhelpful, unsafe, and vulnerable to jailbreak in the presence of conflicting objectives [2, 58]. To build safe and performant foundation models serving a diverse population, we need methods that can explicitly account for and adapt to the inherent plurality of human preferences.

These insights suggest that human preferences are not derived from a single utility function, but are affected by unobserved, hidden user context [58]. To accurately model individual utilities, RLHF should be able to efficiently infer and adapt to the user context. With this in mind, we formulate RLHF as a latent variable problem. Building on techniques from variational inference [11, 38], we propose a method—Variational Preference Learning (VPL)—for multi-modal reward modeling. Intuitively, given a few preference annotations from a particular user, our approach uses a variational encoder to infer a latent distribution over hidden context, and a latent conditional reward model to accurately recover the true *multi-modal* preference distribution. We derive an evidence lower bound (ELBO)

for latent-variable preference-based reward optimization. Our proposed algorithm, VPL, effectively learns a distribution of reward functions from large corpora of preferences from diverse users.

In developing practical training methods for such latent-conditioned reward models, we show that several complexities and technical considerations arise. A key problem is that binary comparisons inherently lack information regarding the scale of rewards. Under the BTL model (and correspondingly the VPL model), the preference label between two alternatives $A$ and $B$ can only provide information about $r_A - r_B$. So, the learned reward function may have vastly varying reward scales across individual users that adversely affect the optimization landscape of multi-user reinforcement learning [31, 69] using these rewards. To mitigate this, we show how a simple pairwise classification scheme [61, 49] can appropriately bound and scale reward estimates within our latent variable framework, thereby enhancing the performance of downstream learned policies. The predicted latent distribution enables several additional capabilities. In Section 4 we use our approach to learn latent-conditioned policies that can personalize to particular users at test time. Additionally, the latent variable reward models can measure uncertainty in the reward distribution [56]. So, in Section 4.2, we use our approach to actively query [8, 10, 9] users to minimize the number of labels they need to provide before we can adapt to their distinct preferences.

Overall, our work introduces a latent variable framework for reward modeling that can encode and approximate the distribution of user preferences directly from binary comparisons, and steer the downstream policy to adapt to diverse user preferences. We conduct a broad range of experiments across three simulated robotics environments and two language tasks with conflicting user preferences. Our results show that in simulated domains, VPL accurately models rewards and improves task performance and personalization. We are able to scale this method to many users, and use active learning to adapt efficiently to particular users with significantly fewer queries at test time. In the language domain, we are able to train multiple LLM-based reward models that learn a separable embedding space that can distinguish between users with divergent preferences. The resulting models outperform existing by RLHF approaches 10-25% [52, 58] by more precisely predicting rewards that align with diverse users and objectives across multiple datasets.

## 2 Related Work

**Reinforcement Learning from Human Feedback (RLHF):** We focus on the problem of reinforcement learning (RL) from binary human preferences using the Bradley-Terry-Luce (BTL) model [15]. This has a rich history in the field of RL and robotics, often referred to as Preference-based RL (PbRL) [65, 28, 1, 8, 10]. We specifically build on the framework outlined in Christiano et al. [17] and further expanded in recent works [52, 2, 74, 60, 37, 58]. This has seen a wide range of applications ranging from training robots [17, 10, 62] to finetuning language models for alignment [52, 74, 60]. Our work, enabling RLHF with diverse preferences, is easily applicable to *any* preference-based learning method, including recent techniques [55, 24] that circumvent reward modeling altogether.

**RLHF under non-BTL models:** Prior work has aimed to study non-BTL models of human behavior and preferences [12, 43, 42, 37, 61] to account for human irrationality and uncertainty in preferences, or intransitive preferences [49, 61, 64]. However, our key argument in this work is less about human irrationality (i.e. inconsistency), and more about the divergence between potentially rational preferences for different labeling users and end users. In this sense, our work is complementary in that the latent variable model can easily be adapted to non-BTL models as well. In fact, we incorporate the technique proposed in Swamy et al. [61] to improve reward learning for downstream applications.

**Personalized RLHF:** While some works [39, 40, 35] characterize similar challenges as VPL, they largely focus on exploring the societal issues underpinning the need for personalization and introducing datasets with diverse annotations. Conitzer et al. [19] argue Social Choice Theory could provide insights into how to aggregate diverse preferences. But these works do not propose a technical method to achieve modeling diverse preferences.

Several prior works have looked at trading off conflicting alignment objectives (such as remaining both helpful and harmless) through techniques like Pareto-optimal optimization [13, 16] or multi-objective RL [67, 21]. For example, Jang et al. [33] treats personalization as a multi-objective RL problem, requiring explicit decomposition into different sub-rewards and learning independent reward models for different users. Further, Dai et al. [21] introduces Safe RLHF, an approach that explicitly models the different objectives of helpfulness vs harmfulness. In contrast, our work does not aim to

reconcile the diversity but rather solve the model misspecification and learn reward models that can infer the context and specialize to a particular user.

Some methods which deal with diverse underlying preferences (e.g. [14]) aim to model disagreement in the data but typically require more information such as annotator demographics [26, 4]. Li et al. [46] learns to map user information to a fixed set of representations for each user, and conditions the reward model on this embedding. Feng et al. [25] proposes a pluralistic alignment framework, using smaller LMs that are trained on community or user-specific data. In contrast, our method does not assume access to personal user information, but only a few preference annotations from each user.

Some works have explored personalization outside of the context of reward learning. Zhao et al. [72] uses in-context learning with few samples from a particular user group, to learn preferences over discrete answers to a question, but does not focus on reward learning with a BTL loss over long context prompt and responses, which is our focus. In addition, our method learns a latent distribution over user types to enable capabilities like active learning [71] and latent-conditioned policy learning.

Dumoulin et al. [23] explores annotator-model misspecification and uses synthetic examples to show how RLHF fails in the presence of two users with different preferences, but they do not propose a scalable approach for resolving this issue. The closest work to ours is Distributional Preference Learning (DPL) [58], which aims to account for hidden context in RLHF. While the motivation is similar, the techniques proposed in DPL include modeling only the mean and variance of the reward distribution across users, or a particular categorical approximation to the reward distribution. As such, DPL can capture uncertainty in the inferred reward distribution, but cannot accurately predict the reward for a particular user. In contrast, VPL explicitly models a user-specific latent variable $z$, and learns $z$-conditioned, user-specific reward models, enabling us to accurately reconstruct divergent reward functions for different users. As we will show in Section 6, this leads to significantly improved empirical reward prediction results. Unlike DPL, VPL takes a variational approach to user modeling and allows for reward interpretability, active learning, and model specialization to users at test time.

## 3   Technical Preliminaries

In this work, we build on the framework of preference-based reward learning (often referred to as RLHF) [17, 1]. In particular, we will consider reward learning methods based on the Bradley-Terry-Luce (BTL) choice model [15]. RLHF has two key phases: (1) inferring a reward function from human-provided labels of ordinal preferences; (2) using reinforcement learning (RL) to train a decision-making policy that maximizes the rewards inferred in step (1). We will instantiate this framework abstractly, which can then be specialized to both LLMs and robotics.

We define a Markov decision process (MDP) [53] $\mathcal{M} = (\mathcal{S}, \mathcal{A}, \mathcal{T}, \gamma, \rho_0)$, with state space $\mathcal{S}$, action space $\mathcal{A}$, transition dynamics $\mathcal{T}$, discount factor $\gamma$ and initial state distribution $\rho_0$. Notably, we do *not* have access to an underlying reward function, but instead have annotators who rank pairs of states $s_A$ and $s_B$. We assume annotators have an unknown reward function $r(s)$ that informs their preference labels. More formally, an annotator ($h \in$ H) takes a pair of states $s_A$ and $s_B$, and returns an ordinal preference i.e. $y = \mathbb{1}(s_A \succ s_B)$, according to $r(s)$ [15, 23]; where H is the space of all possible annotators. Given a dataset of annotated preferences $\mathcal{D} = \{(s_A^i, s_B^i, y = \mathbb{1}(s_A^i \succ s_B^i))\}_{i=1}^N$, a typical RLHF procedure learns a reward function $r_\phi(s)$ using a *maximum likelihood objective* (MLE) on the preferences, where the likelihood: $p_\phi(y \mid s_A, s_B)$ can be defined using the BTL model:

$$p_\phi(y = 1 \mid s_A, s_B) = 1 - p_\phi(y = 0 \mid s_A, s_B) = p_\phi(s_A \succ s_B) = \frac{e^{r_\phi(s_A)}}{e^{r_\phi(s_A)} + e^{r_\phi(s_B)}} \quad (1)$$

Note, that while in this case, the ordinal preferences are on states $s_A$, $s_B$, this can be generalized to trajectories or snippets [17, 52]. Finally, the recovered reward function can then be used to learn (or finetune) a policy $\pi_\theta(a|s)$ that can act to maximize the expected sum of approximated rewards in the environment using standard RL algorithms [57, 30, 41] i.e. $\pi_\theta = \arg\max_\theta \mathbb{E}_{\pi_\theta}\left[\sum_t \gamma^t r_\phi(s_t)\right]$. We note that while the BTL model accounts for some IID noise in preferences through the probabilistic formulation [17, 3, 23], it does not account for hidden-context and divergent human preferences [58], and it does not allow the underlying reward models and policies to be personalized to specific users.

# 4 VPL: Incorporating Latent Context into Preference-Based Learning

The standard BTL formulation described in Section 3 is based on the assumption that all annotators $h \in \mathrm{H}$ share a single underlying reward function $r_\phi(s)$. This does not hold in practice with a diverse range of annotators. To model diverse, pluralistic preferences, we frame multi-modal reward learning as a latent variable problem, where the latent variable $z$ represents the hidden context affecting the underlying reward function (and thereby the preferences) of an annotator $h \in \mathrm{H}$; for instance, it could be representative of their lived experience, or the style of response/policy that they would like the agent to perform. Following this, the latent-conditional reward $r_\phi(s, z)$ is a function of latent $z$ along with state $s$, and preference likelihoods can be expressed with a latent-conditional BTL model:

$$p_\phi(y = 1 \mid s_A, s_B, z) = p_\phi(s_A \succ s_B \mid z) = \frac{e^{r_\phi(s_A, z)}}{e^{r_\phi(s_A, z)} + e^{r_\phi(s_B, z)}} \tag{2}$$

The maximum likelihood objective for this model from a dataset of preference labels (with multiple annotators), $\max_\phi \mathbb{E}_{s_A, s_B, y \sim \mathcal{D}} \left[ \log p_\phi(y \mid s_A, s_B) \right] = \mathbb{E}_{s_A, s_B, y \sim \mathcal{D}} \left[ \log \int p_\phi(y \mid s_A, s_B, z) p(z) dz \right]$ is intractable due to marginalization over the unobserved latent variable $z$. To tackle this, we can introduce a variational posterior approximation $q_\psi(z \mid \{(s_A^i, s_B^i, y^i)\}_{i=1}^N)$, conditional on multiple annotations $\{(s_A^i, s_B^i, y^i = h(s_A^i, s_B^i)\}_{i=1}^N$ provided by the same annotator $h$[2]. This results in a corresponding evidence lower bound (ELBO), $\mathcal{L}(\phi, \psi)$, for the intractable marginal $\log p_\phi(y \mid s_A, s_B)$:

$$\mathbb{E}_{\substack{h \sim \mathrm{H} \\ \{s_A^i, s_B^i, y^i = h(s_A^i, s_B^i)\}_{i=1}^N \sim \mathcal{D} \\ (s_A, s_B, y = h(s_A, s_B)) \sim \mathcal{D}}} \left[ \mathbb{E}_{z \sim q_\psi(z \mid \{(s_A^i, s_B^i, y^i)\}_{i=1}^N)} [\log p_\phi(y \mid s_A, s_B, z)] - D_{\mathrm{KL}}(q_\psi(z \mid \{(s_A^i, s_B^i, y^i)\}_{i=1}^N) \parallel p(z)) \right] \tag{3}$$

This objective samples a user $h \sim \mathrm{H}$ from the given annotators and a set of annotations from this particular user $\{(s_A^i, s_B^i, y^i = h(s_A^i, s_B^i)\}_{i=1}^N$ for inferring the latent variable $z$ through the posterior $q_\psi(z \mid \{(s_A^i, s_B^i, y^i)\}_{i=1}^N)$. Given the posterior, this objective involves optimizing two terms: a maximum preference likelihood objective ($\log p_\phi(y \mid s_A, s_B, z)$) using the contextual BTL model and a regularization term ($D_{\mathrm{KL}}(q_\psi(z \mid \{(s_A^i, s_B^i, y^i)\}_{i=1}^N)) \parallel p(z))$) against a prior $p(z)$. Intuitively this objective encodes a set of user-provided annotations $\{(s_A^i, s_B^i, y^i)\}_{i=1}^N$ into a latent distribution using the encoder $q_\psi$, and then learns a latent-conditional reward function $r(s, z)$ that best explains the annotated preference data. As the variational encoder $q_\psi$ generates a latent distribution, this formulation further enables uncertainty estimation and active learning, as shown in Section 4.2. We describe details further in Appendix B.3.

We emphasize that the only additional requirement of this objective, compared to standard RLHF [17], is a set of annotated pairs from the same annotator, and this information is easily available when annotators are queried in batches [10]. We refer to Algorithm 1 for further details on implementation.

**Personalized, latent-conditioned policies.** The learned reward models $r_\phi(s, z)$ can be used to train personalized, user-specific policies. During training, we learn a latent-conditioned policy $\pi_\theta(\cdot \mid s, z)$ that maximizes the rewards $r_\phi(s, z)$ for different values of $z$. This allows the policy to adapt to diverse user preferences encoded in the latent space. We can use any standard (RL) algorithm [57, 30, 41] to optimize the latent-conditional reward maximization objective: $\pi_\theta = \arg\max_\theta \mathbb{E}_{\pi_\theta, z \in p(z)} \left[ \sum_t \gamma^t r_\phi(s_t, z) \right]$. For this, we sample $z$ from the prior $p(z)$ during training and learn the policy $\pi_\theta(\cdot \mid s, z)$, to maximize the corresponding latent-conditioned reward $r_\phi(s, z)$.

At test-time, we infer a specific user's ($h_{\mathrm{test}}$) latent context $z$ by posterior inference using a set of labeled preference queries $\{(s_A^i, s_B^i, y^i = h_{\mathrm{test}}(s_A^i, s_B^i))\}_{i=1}^N$ i.e $z \sim q_\psi(z \mid \{(s_A^i, s_B^i, y^i)\}_{i=1}^N)$. We then deploy the personalized policy $\pi_\theta(\cdot \mid s, z)$ conditioned on the inferred $z$ for that user. We outline the complete algorithms for policy training and deployment in Appendix C.

## 4.1 Scaled Rewards for Multi-Task Learning

In practice, optimizing latent-conditioned reward functions learned with the VPL objective poses unique challenges. The pairwise preferences used to train the reward model in Section 4 do not have

---

[2]Having multiple annotations are important here to be able to accurately infer the user's latent vector $z$

information about the scale of rewards, but only their relative ordering. As a simple illustration, if we have a pair of states $s_A, s_B$, where the users prefer $s_A$ i.e. $s_A \succ s_B$, two different reward functions: $r(s_A) = 100, r(s_B) = 50$ or $r(s_A) = 50, r(s_B) = 0$ have the same likelihood under the BTL model. Empirically, we observe that this poses problems for optimizing Equation 3; different values of the latent variable $z$ generate reward functions of vastly different scales. This is an issue for several reasons: 1) varying reward scales adversely affect the landscape of multi-user policy optimization (often observed in multi-task RL) [31], and 2) it is challenging to identify states where user preferences diverge across the population as differently scaled rewards cannot be directly compared.

To address this issue, we experiment with several different techniques for scaling the learned reward functions (see Appendix A.2). Our key insight in solving this challenge lies in the observation that while raw rewards from BTL are not scaled equally across $z$, probabilities from the preference likelihood model $p(y \mid s_A, s_B, z)$ are appropriately scaled. This suggests that an effective solution to the reward scaling issue is to replace the raw rewards from the BTL model ($r(s, z)$) with likelihoods suggested by the pairwise preference likelihood model $p(y \mid s_A, s_B, z)$. In particular, a natural choice of scaled rewards for state $s_A$ is the expected likelihood that $s_A$ is "preferred" to all other states (or a sampled set of states) $s_B$ observed in the data - $r_\phi(s_A, z) = \mathbb{E}_{s_B \in \mathcal{S}} [p_\phi(y = 1 \mid s_A, s_B, z)]$. Since these are probabilities, normalized in the $[0, 1]$ range, the scaling of rewards is consistent across latents $z$. Note that these expected likelihood rewards can easily be obtained from the objective in Equation 3 since we train a latent-conditional preference classifier. While proposed from a very different perspective, we note the similarity of this reward scaling approach to recent work [61, 49], in particular, Self-Play Preference Optimization (SPO) [61], which was originally proposed to address the issue of intransitive preferences. Similar to [61] we assume that the oracle providing preference labels is Non-Markovian. Due to this similarity[3], we use VPL-SPO to indicate this approach of preference-based reward scaling throughout our experiments (See Algorithm 3).

## 4.2 Active Learning of Preferences to Minimize Latent Uncertainty

A natural question that arises for test time deployment of the latent-conditioned policies is how to obtain the set of state pairs $\{(s_A^i, s_B^i)\}_{i=1}^N$ to be annotated with preference labels and used for posterior inference, $z \sim q_\psi(z \mid \{(s_A^i, s_B^i, y^i)\}_{i=1}^N)$. Not all query sets $\{(s_A^i, s_B^i)\}_{i=1}^N$ are made equal; some are more informative than others. Certain states (where preferences *vary* across annotators) are particularly informative in accurately inferring the $z$ which should be used for policy deployment $\pi(a|s, z)$. In VPL, the probabilistic modeling of the variational encoder naturally allows for active selection of the most informative query set based on maximal information gain, following prior work [8, 10, 50]. Here the provision of preference labels $\{y^i\}_{i=1}^N$ will provide the maximum information about the latent distribution (and indirectly, the user preferences). This active query selection procedure can be expressed as the following optimization problem, maximizing the mutual information between the labels and the latent distribution.

$$\{(s_A^i, s_B^i)\}_{i=1}^N \leftarrow \underset{\{(s_A^i, s_B^i)\}_{i=1}^N}{\arg\max} \, \mathcal{I}\left(z; \{y^i\}_{i=1}^N \mid q_\psi, \{(s_A^i, s_B^i)\}_{i=1}^N\right) \qquad (4)$$

The posterior $q_\psi$ is a multivariate Gaussian, and assuming a uniform distribution over the set of labels, $q_\psi(z \mid \{(s_A^i, s_B^i)\}_{i=1}^N)$ allows for closed form solution for mutual information $\mathcal{I}$. To solve the maximization objective, we chose the query set $(s_A^i, s_B^i)_{i=1}^N$ with the maximum information gain, across samples from the preference dataset. Finally, we elicit user labels on this maximal query set, infer the latent, and condition the policy on this latent at deployment.

## 5 Scaling VPL for Reward Learning in Large Language Models (LLMs)

VPL can be used to train pluralistic reward models for LLMs, accounting for diverse human preferences and values. Here we discuss the key details that are essential to scale our method to LLMs. The architecture of our LLM reward model is shown in Figure 2. Unlike prior work which attempted to insert summary embedding layers into LLMs (see e.g. [18]), we find that we can successfully compress user preference information into a concise, probabilistic embedding vector $z$ without sacrificing reward model performance. Further details and hyperparameters are discussed in Appendix B.

---

[3]There are some differences in setup with SPO likelihoods being computed against on-policy samples, while VPL-SPO likelihoods are computed against a fixed offline dataset of comparator states.

**Prompt and Response Embeddings.** Since using raw representations of the prompt and responses can increase the context length significantly, we use a pre-trained LLM to encode prompt and response pairs together [6] (to be consistent with previous notation, we assume a preferred state $s^A$ contains both a prompt and response $[x, r]$, and $e^A = LLM(s^A)$). For efficient training, we pre-compute and freeze the encoded embeddings.

**Latent Encoder.** Given a set of multiple encoded preference queries from the same user, $\{(e_i^A, e_i^B, y^i)\}_{i=1}^N$, we pass each through the same pair encoder to obtain $h_i = enc(e_i^A, e_i^B)$. The latent representation $z$ is generated using self-attention layer over the set of encoded pairs, $\{h_i\}_{i=1}^N$.

**Reward learning.** Here, the representation $e^{A'}$ of a new state $s^{A'}$ is concatenated with a $z$ sampled from the posterior distribution which is passed into an MLP to predict the rewards. The LLM is fine-tuned using low-rank adaptation (LoRA) [32], and unlike typical RLHF settings, we find that we need to train the reward model for $\geq 1$ epochs to fit the encoder and the reward model.

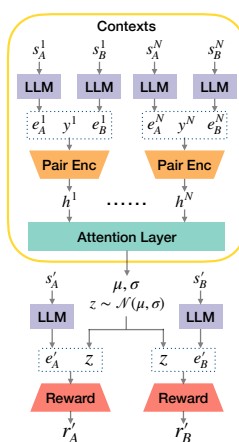

Figure 2: VPL LLM architecture for reward learning. The left and right parts denote the encoder $q_\psi$ and the reward model $r(s, z)$ respectively.

**Data augmentation.** As we scaled VPL to larger datasets with more users, we found that augmenting the training dataset with multiple context samples from the same user for each new data point is essential to learning an effective encoder. Formally, at training time, given a prompt and response pair $s'_A, s'_B$ from a particular user, we generate $M \in \{4, 8\}$ duplicates of this labeled response with different contexts i.e. annotated prompt and response pairs $(\{(s_A^i, s_B^i, y^i)\}_{i=1}^N)_{j=1}^M$; where each context $(\{(s_A^i, s_B^i, y^i)\}_{i=1}^N)_j$ is sampled from a user annotated subset of size $K$ ($K > N$).

## 6 Experimental Evaluation on Simulated Control Tasks

In our experiments, we answer the following questions: (1) Can VPL accurately learn a multi-modal distribution of reward functions from a preference dataset labeled by diverse users? (2) Do the inferred latent user vectors enable learning a multi-task personalized policy? (3) Can we leverage the posterior to actively query preferences for latent estimation? In this section, we show the benefits of VPL in multiple simulated control tasks and demonstrate that VPL is able to capture multi-modality in the underlying reward functions, arising due to underspecification of annotator goals and preferences. In the following section, we ask whether VPL scales up to RLHF for LLMs.

**Training and Evaluation Details:** We test our hypothesis across evaluation domains in two steps: 1) We train a reward model on a dataset of preferences collected using diverse simulated humans; 2) We train a policy using RL to maximize the learned rewards. For these experiments, we use Implicit Q-Learning [41], an offline RL algorithm that achieves strong performance on offline RL benchmarks [27]. Using the learned reward function $r_\phi(s, z)$, and the prior $p(z)$, we label the reward-free offline RL dataset $D = (s_t, a_t, s_{t+1})$, by sampling a latent $z \in p(z)$, and setting the reward as $r_t = r_\phi(s_t)$, or a one-step look-ahead method, where $r_t = r_\phi(s_{t+1})$ (refer to Algorithm 2 for complete method) . We include the hyperparameters and the training details in the Appendix B

**Baselines:** We compare our method against multiple baselines: 1) **Oracle [47]**: This is a goal-conditioned offline RL method, that presents an oracle with access to the true reward functions for all annotators. 2) **BTL [17]**: This is the standard RLHF method from [17, 52] as a baseline, where the reward model is approximated using the *unimodal BTL* function. 3) **DPL [58]**: Following the work on accounting for hidden context in RLHF, we train a distributional reward model, using both the mean-variance (MeanVar) and categorical (Categorical) approximation for the reward functions. 4) **VPL (Ours)**: We denote two versions of our method, **VPL** and **VPL + SPO**, discussed in Section 4.

### 6.1 Tasks:

We evaluate our approach on three diverse simulated control tasks (Figure 4) to demonstrate the effectiveness of VPL to learn latent conditioned policies.

**Maze-Navigation** is based on the D4RL benchmark [27]. Here, users guide a pointmass agent to goals marked with their preferred colors. The user's preferred color is underspecified, so the agent has to infer their preferences from a few annotated comparisons and navigate to one of two/ten locations.

**Ravens-Manipulation** requires the agent to rearrange an object on a table-top (akin to rearranging a dining table), based on user preferences. Built on the ravens benchmark [70], the agent controls a robot arm in a *continuous action space* to pick and place the box in one of the two preferred locations.

**Habitat-Rearrange** [68] tasks a mobile robot arm with relocating a bowl to a user-specified location among five candidates (e.g., desk, kitchen, table). The robot is equipped with motion primitives for navigation and manipulation, and the problem involves reasoning over these locations. This simplified one-step reasoning task highlights the need for personalization in assistive robotics, as each person has distinct preferences for robots in their homes.

**Habitat-Tidy** is a simulated environment similar to TidyBot [66], where the robot infers placement locations for objects in a kitchen based on user preferences. Users may prefer sorting or cleaning objects by different attributes (e.g., function or material), and the robot must adapt to these preferences. Like Habitat-Rearrange, the task involves reasoning over attributes for sorting based on existing preferences, using motion primitives for navigation and manipulation.

### 6.2 Can VPL capture multi-modal reward functions from a dataset of diverse preferences?

We generate preferences using multi-modal reward functions across multiple didactic and simulated discrete and continuous control experiments, as shown in Figures 6, 3, 4, 11 and 10. We see that the BTL baseline with an MLP model averages the different underlying rewards and learns an inaccurate reward model (Figure 3b). In the presence of a majority, BTL converges to the preferences ignoring the minority groups (See Figure 11). While DPL [58] can recover the uncertainty in the reward models due to underspecification, they have no mechanism to recover the individual reward functions. As a result, the DPL reward model estimates high variance rewards (see Figures 6, 10) for each particular user. In contrast, VPL infers the hidden user context using the latent variable formulation and accurately recovers the multi-modal reward distribution.

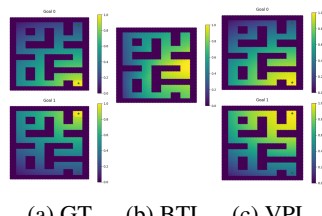

(a) GT   (b) BTL   (c) VPL

Figure 3: Ground truth preferences (a) show that annotators prefer the robot navigate to two different goals. Unimodal BTL (b) averages over the two modes. VPL (c) accurately reconstructs diverse preferences, and learns $z$ conditioned policies to reach either goal.

### 6.3 Do distributional reward functions enable learning a steerable multi-task policy?

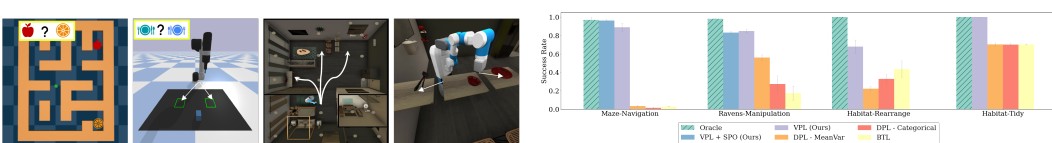

Figure 4: Performance of a downstream policy on diverse control and reasoning tasks, using the rewards trained using different baselines. We report the mean and standard error over five seeds. Note: Habitat envs have a one-step greedy policy so reward scaling and SPO+VPL are not required.

As discussed above, the baselines average the reward modes across the users and learn an inaccurate reward function. So, in Figure 4 the policy for the *Maze-Navigation* converges to a wrong goal, leading to poor performance and misalignment with all the underlying users. For the *Ravens-Manipulation* experiments, the baselines randomly choose a goal location and do not adapt to the user preference at test time. Similarly, for the *Habitat-Rearrange* tasks the baselines are unable to capture the diversity in user preferences over the multiple locations and do not succeed in placing the bowl at the correct location. Finally, in the Habitat-Tidy task, we observe that the baselines converge to an accuracy achieved when they ignore the diverse preferences of sorting the objects by the users.

In contrast, the policies trained using VPL outperform all the baselines in terms of the task success rate, according to the user's underlying reward function. For the navigation task, VPL correctly infers

the goal and the learned policy can navigate to the goals with a high success rate, and the performance is comparable to a goal-conditioned oracle. For *Ravens-Manipulation*, VPL infers the user latent at test time and accurately places the box at the right location. Finally, VPL is mostly able to correctly identify the preferred location for the bowl in the user's home and has a higher success rate than the baselines. We note that scaling the rewards via VPL + SPO improves the performance of multi-task RL for optimizing diverse user preferences. In Habitat-Tidy, VPL is able to infer the user preferences and follow the preferred attributes while placing the objects. However, the accuracy of the robot is dependent on the context length and we discuss this in further detail in Appendix A.4.

### 6.4   Can VPL enable active query selection for latent estimation?

In Section 4.2, we present an objective to actively query users at test time to efficiently infer user preferences. Figure 5 shows that this technique leads to better performance of the learned policy across varying numbers of queries $\|N\|$. This implies that the active learning objective 4 which maximizes information gain over the latent distribution generates queries that are more discriminative and provides a more informative posterior for user identification. This results in a more efficient adaptation of the downstream policy to the distinct user preferences, achieving the same performance with only half the queries. These methods can be potentially transferred to LLMs to query and identify user preferences with minimal questions.

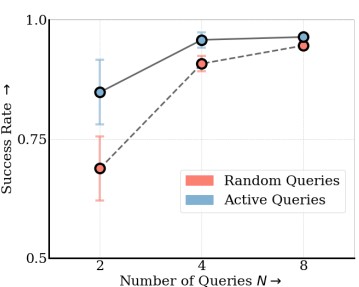

Figure 5: Active learning enables personalizing policies to user preferences with fewer queries.

## 7   LLM Experiments

In this section, we ask: can VPL scale up to the pluralistic alignment of LLM-based reward models? We compare the reward modeling performance of our method against two baselines: the vanilla BTL model and DPL [58]. We experiment with two LLMs: GPT2 [54] and Llama2-7B [63], and two pluralistic preference datasets.

### 7.1   Datasets for pluralistic LLM alignment

Prior RLHF works have focused mainly on unimodal BTL models, and as such there is a lack of publicly available datasets containing annotated preferences with divergent objectives. To evaluate our method on capturing multi-modality in preferences for LLMs, we consider two benchmarks. First, we introduce a synthetic dataset, *Pets*, that directly represents multimodal preferences, and second, we augment the publicly available UltraFeedback [20] dataset.

**Pets.** Here, the dataset is generated to reflect multi-modal user preferences, where each user has a preference ranking over four kinds of animals (in this case cats, dogs, birds, and rabbits). To simulate a setting where users agree on some comparisons and disagree on others, we consider two users who agree on the best and worst pet and disagree on the middle pair of rankings over pets. Preferences here are divergent in certain cases (middle pets), and agree in other instances (best and worst pets), requiring multimodal preference modeling. We evaluate our approach on two versions of the dataset: Pets (Full), and Pets (Divergent) which contains only those prompt and response pairs where the users are divergent (i.e. they have conflicting preferences). For the contexts $\{(s_A^i, s_B^i, y^i)\}_{i=1}^N$, we randomly sample 1-4 other prompts and ranked responses from the same user.

**UltraFeedback-P.** To construct this dataset *UF-P* (where P stands for personalized), we use the fine-grained scores over different attributes available in the UltraFeedback (UF) [20] dataset to construct different users - a similar approach to prior work [58]. We construct a dataset with two users, *UF-P-2*, who prefer either helpfulness or honesty (hidden attribute), i.e. they generate preferences using the scores for their chosen attributes. To test the ability of VPL to model more users than has been previously attempted in the literature, we create *UF-P-4*, which uses the fine-grained scores over all the four attributes in the UF dataset [20] to create a dataset with four different users. Here, the users are divergent because given two responses, users following different objectives can have opposite preferences - some users can prefer the helpful response, while others prefer the honest response. In

Table 1: We compare the accuracy of different reward models trained on the two datasets. We report the mean and standard deviation of performance of GPT2-based models on three seeds, and one seed for Llama models.

| | GPT2 | | | | Llama2-7b |
|---|---|---|---|---|---|
| | Pets (Divergent) | Pets (Full) | UF-P-2 | UF-P-4 | UF-P-2 |
| BTL [52] | $63.27 \pm 0.57$ | $94.92 \pm 0.00$ | $49.84 \pm 0.14$ | $53.48 \pm 0.03$ | 47.17 |
| DPL [58] | $70.62 \pm 1.13$ | $95 \pm 0.00$ | $49.57 \pm 0.42$ | $52.92 \pm 0.06$ | 49.51 |
| VPL (Ours) | $\mathbf{100 \pm 0.00}$ | $\mathbf{100 \pm 0.00}$ | $\mathbf{74.75 \pm 2.01}$ | $\mathbf{61.49 \pm 0.03}$ | $\mathbf{76.41}$ |

some cases, these responses are at odds. To ensure that successfully modeling the data requires fitting divergent preferences, we filtered out the responses where all users agree or are indecisive to primarily focus on multimodal preference modeling, and remove degenerate context queries that provide no information about the user distribution. However, in UF-P-4 the context can still contain queries where at least two users overlap. Thus, this provides a dataset to evaluate VPL in cases where different users agree on some responses, but not all of them. Finally, to generate the context $\{(s_A^i, s_B^i, y^i)\}_{i=1}^N$ for inferring latent distributions, for each prompt and response pair, we sample N different data points from a smaller subset of size $K$ from the dataset ($K = 100$ for GPT2 and 16 for Llama2). For a deployed LLM system, this is analogous to having a known set of survey questions from which the user must answer a subset of 2-8 questions to personalize the model's behavior to their needs.

## 7.2 Does VPL help to make LLM reward models more pluralistically aligned?

In Table 1, we see that VPL is able to learn a more accurate reward model across all the datasets, capturing the multi-modality in the language preference data. This indicates that VPL can infer the latent representation of the user's preferences $z$ from a few annotated samples, and successfully adapt the reward model. In contrast, the baselines—including the BTL model typically used in widely deployed RLHF [52] models—are unable to fit the datasets because they do not account for divergent preferences. Because the datasets are imbalanced, the baselines can sometimes perform better than random guessing by fitting only the preferences of the majority group, and thus, the performance on Pets (Full) appears high, even though the baselines fail to adapt to the divergent preferences.

## 8 Conclusion

In this work, we presented VPL, a technique for pluralistic alignment of preference-based RLHF models via variational inference. We show that VPL can capture diverse preferences and be used for steerable personalized model learning while capturing uncertainty in preferences. We discussed practical considerations for scaling VPL for LLMs and policy learning and showed results across simulated control problems and LLM-based RLHF, significantly outperforming current RLHF techniques.

**Limitations and Future Work.** A key limitation of this work is that as yet, realistic preference datasets containing the opinions of diverse users do not yet exist at scale. This limitation necessitated creating our synthetic preference datasets. Although this was also the approach taken in prior work on personalized RLHF (e.g. [58, 73]), an important direction for future work will be to apply VPL to more realistic preference data from diverse groups of users. Further, our current experiments on the UltraFeedback dataset assume that when adapting to a new user, we could ask them for preferences over samples from a fixed set of survey questions. In the future, it would be useful to relax this assumption and apply VPL to preferences obtained naturally during a conversation with the user.

We believe VPL could offer promising safety benefits in addition to modeling diverse user preferences. Uncertainty detection may help prevent jailbreak attacks caused by conflicting rewards [58]. By capturing uncertainty in the distribution of user preferences, VPL could enhance safety by stopping or refusing to respond when uncertainty cannot be sufficiently reduced [36].

## 9 Acknowledgement

We would like to thank members of the WEIRD and Social RL for the valuable discussions and feedback on the project and manuscript. This project was partly supported by NSF grant no. 2212310.

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

# A  Additional Experiments

## A.1   Didactic example on a toy reward learning problem.

To more carefully understand the behavior of VPL empirically, we construct a didactic example [23] as shown in Figure 6. In this problem, let us consider a mixture of $M$ different annotators providing preferences, where each annotator $i$ has a reward function specified by $\mathcal{N}(\mu_i, \sigma_i)_{i=1}^M$ that they use to assign binary preferences. Mathematically, we sample the preferences from a mixture of Gaussians:

$$p(s_A \succ s_B \mid i) = \frac{e^{r_i(s_A)}}{e^{r_i(s_A)} + e^{r_i(s_B)}}; \text{ where } e^{r_i} \sim \frac{1}{\sigma_i \sqrt{2\pi}} e^{-\frac{1}{2}\left(\frac{x - \mu_i}{\sigma_i}\right)^2}$$

Multi-annotator preferences are simulated by sampling an annotator from this mixture distribution and then assigning binary preferences according to the chosen reward function. We train VPL as described in Section 4 to recover the underlying distribution over reward functions. As expected in Figure 6, standard RLHF [17] averages over the different modes since it can only represent a single reward function. While prior work in accounting for hidden context in RLHF (DPL [58]) can learn the uncertainty in the reward functions due to hidden context, it is not able to accurately disambiguate different modes. Meanwhile, VPL is able to infer the underlying context using the approximate posterior $q_\psi$ and the recover the individual reward modes through the latent-conditional reward function $r(s, z)$.

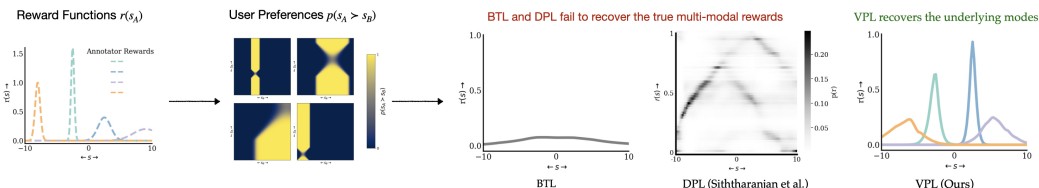

Figure 6: Didactic experiments comparing standard BTL [17], DPL [58] and VPL (Ours). Four Gaussian reward functions generate different binary preference data. The traditional BTL approach [17] averages the different modes, and DPL [58] captures the uncertainty in the rewards due to the multi-modality but cannot accurately predict the true modes. VPL (ours) infers the hidden latent as described in Section 4 and recovers the individual distribution of reward functions.

## A.2   Does scaling rewards help improve performance?

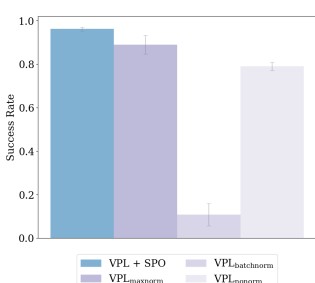

Figure 7: Comparing scaling methods on Maze-Navigation.

To avoid the problem of high variance rewards (Section 4), we compare the performance of $\text{VPL}_{\text{no-norm}}$ with VPL + SPO. We further compare against two normalizing schemes: $\text{VPL}_{batchnorm}$ where the rewards for each latent is normalized by the mean rewards across a set of state samples i.e. $r'(s, z) = \frac{r(s,z)}{\frac{1}{M} \sum_{s' \in \mathcal{S}} r(s', z)}$, and $\text{VPL}_{\text{max-norm}}$ where all the rewards in the offline dataset are normalized by the maximum reward for any latent.

In Section 4.1, we discuss the problem of generating scaled rewards from latent variable-based reward models and compare the performance across multiple baselines discussed above. As shown in Figure 7, the batch norm scaling generates highly biased estimates of the rewards, which is catastrophic for the method. However, VPL methods have decent performance at test-time, but are an unprincipled approach to the scaling problem. Our SPO + VPL presents a general method for estimating normalized rewards. Thus, in Figure 7 we can see that our method outperforms the baseline approaches in terms of success rate. The baselines have an unscaled or a biased estimate of the multi-modal rewards leading to sub-optimal performance. For the ravens-manipulation environment, the dataset doesn't contain sub-optimal trajectories, VPL (with max norm) performs comparably.

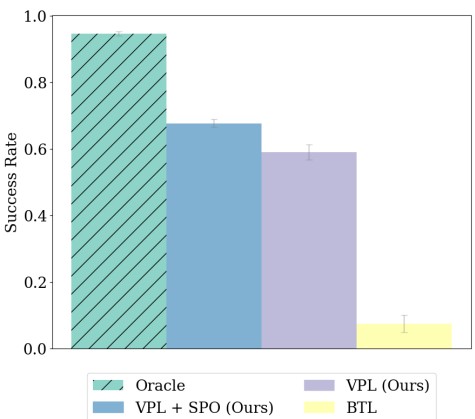

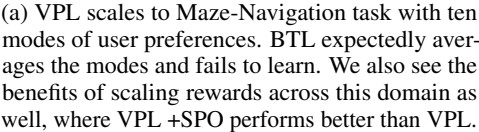

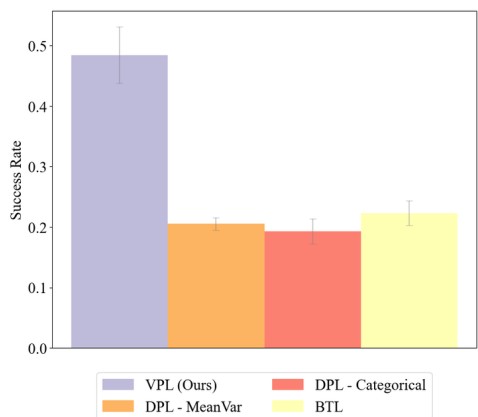

(a) VPL scales to Maze-Navigation task with ten modes of user preferences. BTL expectedly averages the modes and fails to learn. We also see the benefits of scaling rewards across this domain as well, where VPL +SPO performs better than VPL.

(b) We compare the performance of baselines and VPL on a Habitat-Rearrange environment with 100 users. VPL can scale to a much larger set of diverse users, complementing the real-world capabilities shown in Table 1 and Figure 4.

Figure 8: Comparison of VPL's scalability on different tasks and user bases.

## A.3 Does VPL scale with the number of diverse users?

In order to test the effectiveness of VPL in scaling to a control problem with larger modes of underlying preferences, we create a task with ten underlying locations that the users could prefer. So, the challenge here is to disambiguate the user preference among a larger space of possible goals and condition the policy to navigate successfully to the goal. Figure 8a shows that our method is able to navigate to the individual goals with a higher success rate, whereas the baseline DPL model [17] collapses to a single user mode. This demonstrates the benefits of scaling VPL to a setting with a large population of diverse users. To test the method at a larger scale, we increase the number of users in the Habitat-Rearrange tasks to 100. It is a combinatorial problem as the users provide rankings over five locations, so all possible users/orderings are 5!. We observe in Figure 8b that VPL significantly outperforms the baselines in inferring the user preference, and steering the robot policy accordingly.

## A.4 How does context length affect VPL?

In Habitat-TidyBot, the robot's task is to relocate an object in the kitchen (fork, knife, spoon, bowl, pitcher) to a specific location based on user preferences. The user can prefer to arrange objects according to the object function (i.e. kitchenware or tableware), or material (metal or plastic). The agent has to query the location of some of the objects, infer the user type, and arrange the requested object accordingly. Fig 4 shows that the baselines converge to the correct location for the objects agreed upon by the different users, but converge to one mode for the divergent choices. Meanwhile, VPL infers and adapts to user preferences and aligns with humans perfectly. However, one caveat to this is the context length i.e. the number of queries to each user. In Figure 9 we show that as the query length increases VPL can identify users with higher accuracy and achieve higher performance. This happens because certain queries

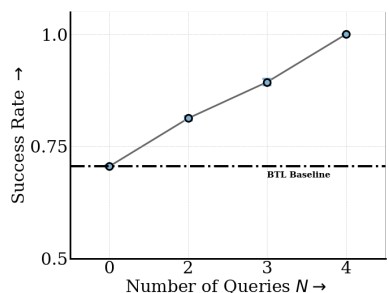

Figure 9: In the Habitat-TidyBot tasks, the agent has to relocate objects in the house based on the user's preferences. We show that the accuracy of user modeling and choosing the right location for the object increases with the number of queries the agent makes to infer the latent distribution.

are uninformative about the user preferences (such as things users agree on), and thus, it generates a high variance posterior. As a result, longer context length increases the probability of useful queries,

which enables low variance posterior inference and improved alignment during decision-making. In Section 4.2, we also show how to use an active learning approach to generate queries based on a max information gain objective. For LLMs, in Figure 13, we show that a higher context length also provides robustness to noise in the annotated queries.

## A.5 Visualizing Learned Rewards and Embeddings

We visualize the rewards generated by the baseline and the latent conditioned rewards models on the diverse domains described in Section 6. We observe that VPL reconstructs the multi-modal reward functions, based on the inferred latent distribution. Figure 3 shows that the BTL models averages the reward over the user-preferred goals, while VPL accurately reconstructs the individual user-specific rewards. Figure 10 shows that for optimal trajectories solving the task, VPL can accurately match the ground rewards for the two modes. At the same time, DPL [58] predicts high-variance rewards for both cases due to the inherent multi-modality. Finally, in Figure 11 we see that the majority of users consider the desk to be the preferred location of the bowl, and standard BTL models converge to the majority population. Meanwhile, VPL can generate user-specific rewards, satisfying all the user groups.

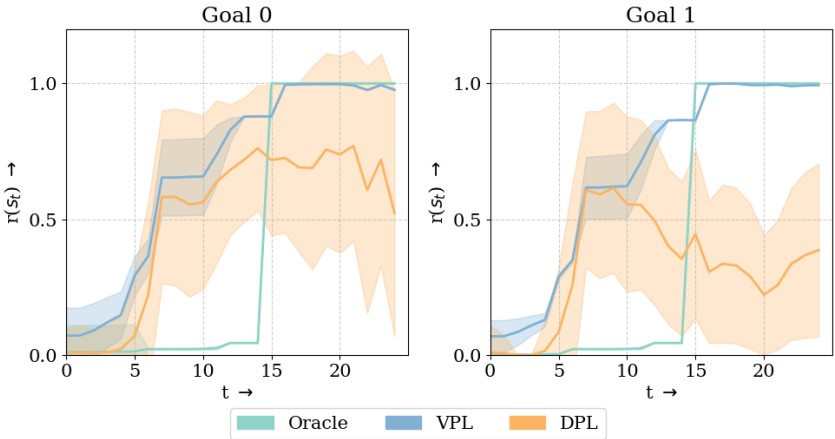

Figure 10: In the Ravens-Manipulation task, we compare the predicted rewards for states $s_t$ along timesteps $t$ in oracle trajectories to either of the goals the user prefers. VPL (Ours) can learn the individual reward functions for the two different (closely matching the ground truth rewards for both users) leading to more performant policies (see Figure 4), while DPL [58] learns a high variance reward function due to the multi-modality.

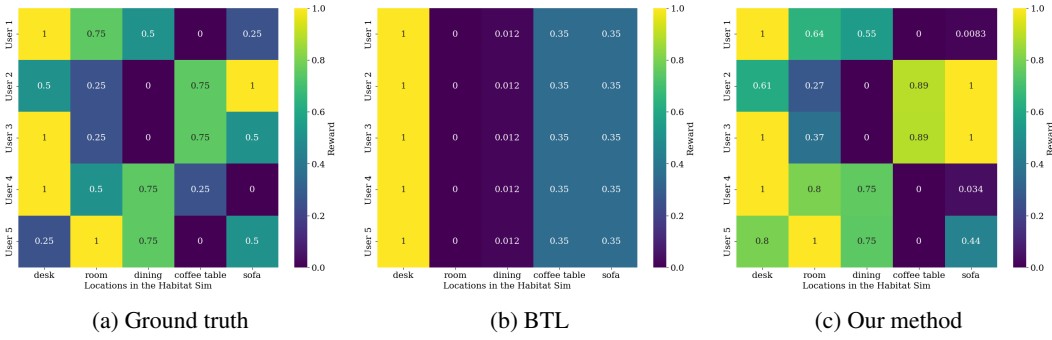

(a) Ground truth          (b) BTL          (c) Our method

Figure 11: In the Habitat-Rearrange task, annotators have rankings, i.e. preferences over the different locations they want the robot to place their bowl in their home. Accordingly, (a) shows the rewards associated with a particular location ("column") for each annotator ("row"). We see that a majority of the users rank the desk as the most preferred location. Consequently, in (b), unimodal BTL converges to this majority preference ignoring other users. However, in (c) we see that VPL accurately reconstructs diverse preferences and aligns to all five users.

## A.6 Analysing the Latent Space

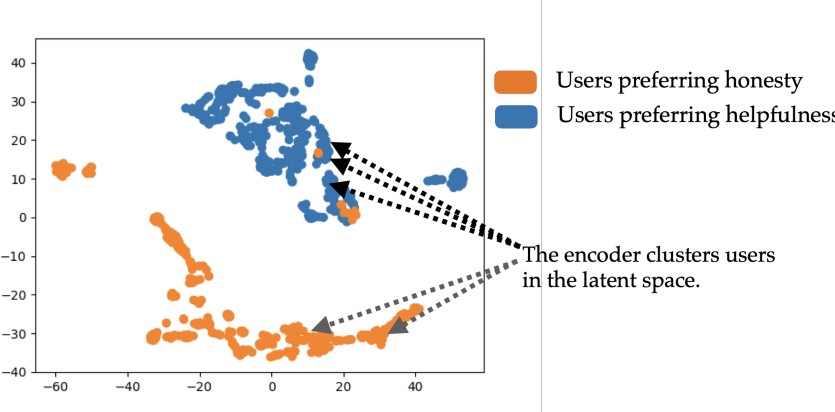

Figure 12: We train GPT2-based VPL, on the UltraFeedback-P dataset. In this plot, we visualize the T-SNE features of the latent distribution $z$ produced by the encoder $q_\psi$ on a set of annotated prompts and responses $\{s_A^i, s_B^i, y^i\}_{i=1}^N$ from the two users in the dataset. We see that the encoder clusters the users in the latent space, allowing the decoder to personalize the reward models according to multiple objectives preferred by the diverse users belonging to a cluster.

In Figure 12, we show that the VPL-Encoder effectively learns a latent space with clusters that correspond to the user types in the dataset. Previous work has shown that attempting to compress information within an LLM into a single bottleneck embedding layer can hinder performance [18]. However, using the architectural design of VPL as well as user-context data augmentation, VPL is able to learn a compressed user representation that accurately separates users according to their preferences, from only a few preference labels.

## A.7 Does VPL scale to real-world settings with larger and noisy users?

A key assumption in our approach is that context questions accurately represent individual users without noise in the underlying dataset. To test VPL's robustness to noisy context labels at test time, we injected noise by flipping the questions answered by each user and evaluated the trained model's accuracy in predicting rewards. This experiment can help us evaluate how well the model would generalize to new users that have similar preferences to those experienced during training, but may not answer questions in exactly the same way. Figure 13 illustrates that VPL is able to outperform prior work even when 25% of preference labels are flipped at test time. Notably, we observed that longer context lengths result in more accurate reward modeling, even with higher noise injection. This is because the encoder can generate more accurate inferences of the latent distribution when provided with more user information through a larger number of context questions and response labels.

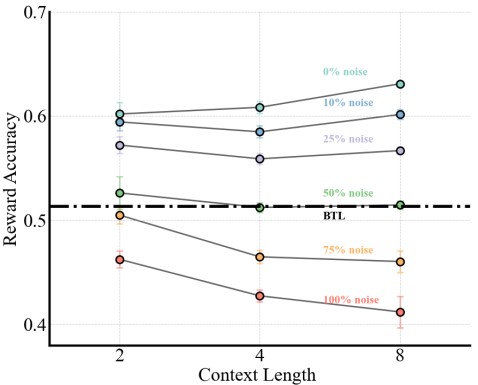

Figure 13: Reward accuracy with varying levels of noisy labels introduced at test time.

We note further that when 50% of the preference labels are flipped, the context preference queries provide no information about the user, and the performance of VPL is equivalent to the baseline BTL model. This mirrors the additional findings presented in Appendix A.8, which demonstrate that when VPL is trained on a unimodal preference language dataset, it gives equivalent performance to BTL. Essentially, when extra preference data is available for a user, VPL can personalize the reward model effectively and attain higher performance. But without that information, it performs just as well as the

default BTL model. These findings, as shown in Table 1,Figure 13, and the Appendix, demonstrate VPL's effectiveness in handling multi-modality and noise in large-scale preference datasets, without compromising performance even when no additional preference information is available. This capability suggests the potential for VPL to contribute to the development of next-generation LLMs that are more personalized, inclusive, and efficient.

### A.8 Does VPL work under uni-modal settings?

To test that the introduction of a variational framework does not decrease performance in settings where all users have single/aligned preferences, we run an experiment on the UF-P-4 dataset, where we considered the preferences of a single user (preferring the model to be "honest" over all other attributes) to analyze the single-modal case as suggested. The standard BTL model gives a 77.04% eval accuracy while our VPL model gives a 77.16% eval accuracy. Our model matches the baseline performance, indicating no drop in performance when using VPL compared to traditional RLHF over an unimodal dataset.

### A.9 Social Impact

This work has a clear social impact when deployed on user-facing systems like LLMs or household robots. In pluralistic alignment, we assume that some differences in user preferences reflect divergent but equally valid perspectives on which moral or cultural values an LLM should align to; for example, individuals from one culture may hold collectivist moral values, while another culture is more individualist. Both value systems should be respected, and as such LLMs should be able to recognize and adapt to the values held by a particular user. However, the personalized model could potentially either be sycophantic or align with adversarial users, which is undesirable. This raises very interesting questions, such as: At what point should the LLM embrace a more universal set of values? How can we detect when such a point has occurred in a particular conversation? The probabilistic framework of the user distribution could allow us to identify low probability or rare behavior, and also the distributional nature of reward functions can help us point out responses where the users are divergent (maybe signifying disagreement). Additionally, a model could flexibly switch between adhering to the user's personal preferences and conforming to a more universal perspective on topics where it could be biased, or is sensitive to jailbreak [2]. Taking inspiration from Constitutional AI [2], we can allow a system designer to specify the topics for which the LLM should not engage in user personalization. Overall, this presents an exciting future research direction toward building safe personalized LLMs.

## B Implementation Details

### B.1 Task Details

We evaluate our methods on three simulated control environments.

**Maze-Navigation.** This task is adapted from the "maze2d-medium-v2" environment from the D4RL benchmark [27]. The observation space is the position and velocity of the robot $(p_x, p_y, p_z)$, and the pointmass is controlled using torque control. In this environment, point mass doesn't have access to a goal, and diverse users guide the agent to (two or ten) different locations in the maze, marked with their preferred colors. The users label the preferences over two states based on the shortest path to the goal from each state, i.e, the user prefers states closer to their preferred color location. The offline dataset for IQL is collected using the waypoint controllers provided in the D4RL benchmark. For each episode, the agent is spawned at a random location in the maze, interacts with a random user, and navigates to the goal based on the learned reward model and corresponding policy trained on the offline dataset. The oracle reward function is the optimal Q-value of the state, generated using a dynamic programming solution, which is available in D4RL.

**Ravens-Manipulation** This task is adapted from the ravens benchmark [70]. The observation space is the 3-D position of the object, the 3-D position of the end effector, and the grasp state of the object, i.e., $(ee_x, ee_y, ee_z, p_x, p_y, p_z, grasp)$. The agent commands absolute positions for the 3-DOF robot arm in end-effector space i.e. $(ee'_x, ee'_y, ee'_z)$. This setup resembles how a robot arm would

infer user preferences to organize a dining table. The users prefer two different locations for the box spawned at a random location at the beginning of each episode. To collect offline data, we use a motion planning oracle with some added noise, which tries to pick the box and place it randomly at one of the two locations. The oracle reward function is as follows:

$$\text{reward} = \frac{1}{100} \begin{cases} 100 & \text{if goal\_dist} < 0.05 \text{ and not grasped} \\ 5 & \text{if goal\_dist} < 0.05 \text{ and grasped} \\ 2 + \exp(-\text{goal\_dist}) & \text{if not goal\_dist} < 0.05 \text{ and grasped} \\ \exp(-\text{gripper\_dist}) & \text{if not goal\_dist} < 0.05 \text{ and not grasped} \end{cases}$$

**Habitat-Rearrange**  This is a task based on the Meta Habitat simulator [68]. Here, the objective for the Mobile Manipulator is to pick a bowl and place it at the user's preferred location in the home. However, the exact location is underspecified and needs to be inferred from the user-annotated preferences. The robot uses a motion primitive to navigate and place the object at five possible locations ('desk', 'room', 'dining', 'coffee table', 'sofa'). This problem is reduced to a discrete one-step problem, where the robot has to reason about the best possible location to put the bowl. For ranking the states, the users are generated by randomly choosing a five random orderings of the locations, each corresponding to an individual user. At test time, the agent is greedy and chooses the location with the maximum inferred reward from the learned reward model.

**Habitat-TidyBot**  This task is based on Meta Habitat [68] and inspired from the TidyBot task [66]. In this environment, there are 5 objects in the kitchen (spoon, knife, plate, bowl, and spatula). Each user has their preferences for sorting the objects according to particular attributes (material such as metal, plastic or function i.e. tableware or cooking ware). The robot observes or queries the user for the existing location of a subset of objects, then rearranges the misplaced object according to the inferred user preference and greedily selects the goal with the higher reward. The baseline here would converge to sort the objects according to one attribute only, while VPL would infer the latent, and choose the correct location for the given object.

### B.2   LLM Preference Learning Dataset Descriptions

**Pets**  This dataset is generated synthetically to specifically study the ability of models to perform divergent preference modeling. The goal here is to choose between different types of pets. For each animal, including bird, cat, dog, and rabbit, we use GPT-4 to generate 100 sentences that describe these kinds of pets. Then we define two user groups based on their preference order over the pets. So as to have mix of contexts where users agree and disagree, we construct a preference ordering where both groups like birds the most and rabbits the least. One group of users prefers dogs to cats while another group disagrees and prefers cats to dogs. That is to say, among all 6 comparisons between two kinds of pets, only one pair (dogs versus cats) leads to divergent opinions, while the users agree on other comparisons (birds better than dogs, dogs better than rabbits and so on). This tests the ability of the preference models to capture multimodality, even when the users do agree on some set of preferences.

We then construct the Pets dataset by clustering the prompt and ranked responses according to the group of preferences that they align with. To generate the Pets dataset, we randomly sample a pair of different pets as well as two corresponding sentences, and then label them to be "chosen" and "rejected" according to the preference of either "dog group" or "cat group". The prompt is fixed to be "*Human: Please talk about one kind of pet.*" After all the chosen/rejected pairs are generated, we randomly sample 1-4 pairs of responses from the same user that are comparing dogs and cats for use as the context to the variational encoder. These are informative since a user's choice over these contexts can clearly express the user's preference group. In this way, we can generate a "full" Pets dataset, and based on that we filter out a "divergent" split that only contains controversial data points (comparing dogs and cats). This dataset is meant as a didactic test for language modeling capabilities, but scalability is further tested in the following section with the UltraFeedback-P dataset.

Here we show an example data point for Pets.

- Prompt: "Human: Please talk about one kind of pets."
- Response A: "Cats communicate through vocalizations." (Rejected)

- Response B: "Birds exhibit complex social behaviors within flocks." (Chosen)

- Contexts: ["chosen": "Cats have a preference for certain types of litter.","rejected":"Dogs enjoy exploring their surroundings."], ["chosen":"Cats have a preferred scratching substrate.","rejected": "Dogs have a unique personality."]

**UltraFeedback-P**    The original UltraFeedback dataset [20] contains 64,000 pairs of responses, where each response is evaluated across four dimensions: helpfulness, honesty, instruction following, and truthfulness. Instead of using the averaged scores, we leverage the fine-grained scores to create a dataset with multi-modal, divergent user preferences. We assume that each user ranks responses based on the score for only one attribute. For example, one user might prioritize helpfulness, while another values honesty the most. Consequently, the same response pair can receive divergent labels from different users depending on their chosen attribute. Following prior work [2, 58], we only consider the first two attributes: helpfulness and honesty, to create the dataset UF-P-2. Next, we filter out response pairs where the two users agree or are indecisive, since we care about multimodal reward modeling in this work, leaving approximately 4,000 prompt and response pairs labeled by each user. From this dataset, we randomly sample a smaller subset of K data points, which serves as our context sample set to help identify the latents corresponding to a particular user. For each prompt and pair of responses, we get 8 samples from the context set to characterize the particular user providing responses. These samples, along with the prompt, responses (with preferences over which response is chosen), form a complete data point (context + prompt + responses). This data point is then used to infer the latent distribution and personalized reward for that specific user. Additionally, we introduce a version of the dataset UF-P-4, where we consider all four attributes as distinct users. In the filtering step, we first filter out the response pairs where all of the four users agree on. Then within the data subset of each user, we filter out the response pairs where that particular user gives equal ratings to the two responses. It results in a dataset of approximately 7,500 prompt and response pairs labeled by each user. The way we generate the context is similar to UF-P-2, except that we randomly pick a number from 2 to 8 to be the number of samples to form the context. Note that in UF-P-4, it is still possible that a question from the context set cannot help with distinguish between two users, because we only filter out the data points that are agreed by all users. This presents a large, diverse and challenging benchmark for pluralistic alignment, created from the existing available open-source datasets [20].

### B.3    Implementation Details

**Learned Prior.**    While the methods described in the methods section only learn the reward model and the latent encoder, we can also use a prior $p(z)$ as is common in variational inference methods. We assume that our prior is a multi-variate Gaussian with mean $\mu$ and covariance $\Sigma = \text{diag}(\sigma\sigma^T)$, where $\mu, \sigma \in \mathrm{R}^d$; where $d$ is the dimension of the latent embedding. In all experiments, they are initialized from a standard Gaussian. However, in our control experiments, we observed that using a learned Gaussian i.e. setting $\mu$ and $\sigma$ to learnable parameters under the ELBO objective in Eq. 3 improved performance and stability during training.

**LLM Embeddings.**    In our experiments, we use the embedding from the last token as our encoding for the prompt+response input. However, we also tried llm2vec [5] and a weighted pooling mechanism [48]. However, we find that using the last token embedding as inputs to the encoder and for predicting the rewards performs the best.

**Active Learning Complexity.**    In our active inference technique, we use a sampling-based method inspired by [8] to generate the active queries for the model. Given a dataset of D queries $(s_A^i, s_B^i)_{i=1}^{|K|}$, we sample $S$ query batches of size $Q$, where $Q$ is number of annotations per batch we get from a user (total possible combinations are $^K C_N$). Here, $Q \in [2, 8]$, so we need to perform O(S * Q) passes over the model with batch size $2^Q \sim [4, 256]$. Furthermore, this process only needs to be performed once after the model is trained to obtain the most discriminative set of queries for the given model. Finally, whenever a new user interacts with the system, we need to get labels on the actively inferred queries (usually 2-4) but do not require any additional passes over the query dataset. In our experiments (Figure 5), we show that using active learning allows the model to achieve comparable performance with fewer queries ($\sim 2$), as compared to randomly sampled larger ($\sim 8$) queries.

## B.4 Hyperparameters

Table 2: Hyperparameters for learning reward models using VPL. We sweep over these values and report the best results on 5 seeds.

| Hyperparameter | Value |
|---|---|
| Encoder and Decoder Architecture | MLP |
| Hidden layers | 2 layers of width 256 |
| Optimizer | Adam (Kingma & Ba, 2015) |
| Learning rate | $3.0000 \times 10^{-4}$ |
| Latent dimension | $\{8, 16, 32\}$ |
| $\beta$ | Cosine annealing between 0 and 1 every 25% steps |
| VAE Prior | Multi-variate gaussian with learnable parameters $\mu, \sigma \in \mathcal{R}^d$ |
| Context set of queries $\|N\|$ | 2,4,8,16 |
| Comparison set size (for SPO + VPL) | 1000 |
| Number of annotated sets | 5000 (Maze), 10000(Ravens) |

Table 3: Hyperparameters for IQL. We use the same parameters across all experiments.

| Hyperparameter | Value |
|---|---|
| Architecture | MLP |
| Hidden layers | 2 layers of width 256 (4 layers of width 1024 for users > 5) |
| Optimizer | Adam (Kingma & Ba, 2015) |
| Learning rate | $3.0000 \times 10^{-4}$ |
| Discount | 0.99 |
| Expectile | 0.9 |
| Temperature | 10 |
| Dataset size | 4M steps (Navigation), 5K trajectories (Manipulation) |

Table 4: Hyperparameters for LLM experiments

| Hyperparameter | Value |
|---|---|
| Pair Encoder Architecture | 2 layer MLP with LeakyReLU |
| Hidden Dimension | 512 (GPT2), 1024 (Llama2-7b) |
| Latent Dimension | 512 (GPT2), 1024 (Llama2-7b) |
| Learning rate | $1.0000 \times 10^{-4}$ |
| Learning rate scheduler | Cosine with 3% warmup steps |
| Context set size $N$ | 8 |
| Full context sampling set $K$ | 100 (GPT2), 16 (Llama2-7b) |
| Batch size | 32 (GPT2), 512 (Llama) |
| Optimizer | AdamW(with weight decay = 0.001) |
| $\beta$ | 0.0001 (for Pets), 0.0 (for UltraFeedback-P) |
| Computational Resources | $2 \times$ RTX4090, $4 \times$ A100 |

# C Algorithms

---

**Algorithm 1** Learning Multimodal Reward Functions using VPL

---

**Require:** Preference Data $\{(s_A^i, s_B^i, y^i)\}_{i=1}^N$
**Require:** Encoder E, Reward Model R, prior $p(z)$
1: **while** not done **do**
2:     Sample a batch $B \sim D$
3:     Compute $\mu_B, \sigma_B = \mathrm{E}(B)$
4:     Sample $z \sim \mathcal{N}(\mu_B, \sigma_B)$
5:     Append $z$ to $B$: $\{(s_A, s_B, y)\} \rightarrow \{((s_A|z, s_B|z), y)\}$
6:     Compute rewards: $r_{s_A} = \mathrm{R}(s_A|z)$ and $r_{s_B} = \mathrm{R}(s_B|z)$
7:     Compute reconstruction loss: $\mathcal{L}_{\mathrm{recon}} = \mathrm{cross\ entropy}(\sigma(r_{s_A} - r_{s_B}), y)$
8:     Compute KL-loss: $\mathcal{L}_{KL} = \beta \cdot D_{KL}(\mathcal{N}(\mu_B, \sum_B) \parallel p(z))$
9:     Compute total loss: $\mathcal{L}_{\mathrm{total}} = \mathcal{L}_{\mathrm{recon}} + \mathcal{L}_{KL}$
10:    Update E and R by optimizing $\mathcal{L}_{\mathrm{total}}$
11: **end while**

---

---

**Algorithm 2** Policy Optimization using IQL and VPL

---

**Require:** Offline Dataset $\{\tau_1, \tau_2, \dots\}$
**Require:** Reward Model $r_\phi(s, z)$
**Require:** Prior $p(z)$
**Require:** Policy $\pi(a|s, z)$
1: **for** each trajectory $\tau_i = \{(s_t, a_t, s_{t+1})\}_{t=1}^T$ in $D$ **do**
2:     Sample $z \sim p(z)$
3:     **for** each state $s_t$ in $\tau_i$ **do**
4:        Compute reward $r_t = r_\phi(s_t, z)$                # Alternatively, $r_t = r_\phi(s_{t+1}, z)$
5:        Update dataset with $(s_t, r_t, z)$
6:     **end for**
7: **end for**
8: Train policy $\pi(a|s, z)$ using IQL

---

---

**Algorithm 3** Policy Optimization using IQL and SPO + VPL (Note the changes from Algorithm 2)

---

**Require:** Offline Dataset $\{\tau_1, \tau_2, \dots\}$
**Require:** Preference Model $p_\phi(s_A, s_B, z)$
**Require:** Prior $p(z)$
**Require:** Policy $\pi(a|s, z)$
**Require:** Comparison set $C = \{s_1, s_2, \dots, s_C\}$     # Sampled randomly from the offline dataset
1: **for** each trajectory $\tau_i = \{(s_t, a_t, s_{t+1})\}_{t=1}^T$ in $D$ **do**
2:     Sample $z \sim p(z)$
3:     **for** each state $s_t$ in $\tau_i$ **do**
4:        Compute reward $r_t = \frac{1}{\|C\|} \sum_{s' \in C} p_\phi(s_t, s', z)$
5:        Update dataset with $(s_t, r_t, z)$
6:     **end for**
7: **end for**
8: Train policy $\pi(a|s, z)$ using IQL

---

