# OpenReview forum: "Personalizing Reinforcement Learning from Human Feedback with Variational Preference Learning"
_NeurIPS.cc/2024/Conference — NeurIPS 2024 spotlight_

### Official Review · Reviewer_hKQW · 2024-06-26

**Soundness:** 3
**Presentation:** 3
**Contribution:** 3
**Rating:** 6
**Confidence:** 4

**Summary:**

This paper approaches the challenge of pluralistic alignment, which arises when the preferences among humans diverge across a population. The work identifies that current preference modeling assumptions do not account for multi-modal reward distributions, which is often the case for pluralistic preferences. Thus, they end up averaging modes and generating inaccurate reward models. To address that, the paper proposes a latent variable model that explicitly models different users and applies variational inference techniques to infer the user latent that conditions the reward model and the optimized policy. The paper also proposes scaling rewards following the likelihood from the model, as a way to ensure all latents follow the same scale, which is necessary for latent-conditioned policy optimization. Finally, the paper evaluates the proposed model in a set of control and language environments, demonstrating gains over prior preference modeling choices.

**Strengths:**

- The paper approaches pluralistic alignment, a challenge that is mostly always overlooked by the RLHF community, which implicitly assumes that all humans present the same preferences and values. Therefore, the work is very relevant;

- The main merit of the paper is to identify that current modeling assumptions, while assuming the same preference over humans, fall short when modeling diverse preferences. The paper brings very didactic cases that explicitly show such failure and the consequences in the optimized policy (e.g., Figure 3).

- The introduction of the latent variable modeling for preference modeling in RLHF is interesting and novel, and a natural extension of the prior work [1].

**Weaknesses:**

- The major concern is on the evaluation setup. Although the paper brings a diverse set of experiments, they are all setups where the hidden context is composed of very few variables with simple distributions. This is acceptable to show the failure of previous methods, but not sufficient to claim scalability in the LLM setup for the proposed method. In more detail:
    - The “Pets” dataset is too simple and does not even require language. From the example in the Appendix B.2, it is possible to ignore the prompt and extract the pet variable from the responses and context. The problem then boils down to fitting a categorical classifier over four classes, where the input is the pet in response A and B, and the same for the context. Given the amount of variables and the dataset size, it is questionable if variational inference is required here - perhaps a simple linear model or MLP can learn directly from the context.
    - The “Ultrafeedback-P” indeed contains much richer prompts and responses, and the hidden variables (helpfuness, honesty) are not straightforward to extract from the responses. However, the experiment assumes only two users, and there are only two hidden variables as well. Again, this raises concerns if variational inference is required here - perhaps a base LLM can already separate these users in the feature space, but there is no such a baseline in the work.
        - Interestingly, the paper highlights the requirement of filtering out the context where the users agree, which simplifies the task, as it is easier to identify users with extreme disagreement. It would be important to justify why that is necessary and if it is not a limitation of the proposed method.
    - The paper is motivated by personalized RLHF and provides modeling for latent-conditioned policies, but in the LLM experiments there is only preference modeling and no RLHF/alignment. This time of experiment is necessary to claim that VPL scales up for pluralistic alignment.
    - Some methodology details are missing: how the prior is learned (the training objective, data); how humans are simulated in the control environments (L279); how the accuracy of the reward model in Table 1 is defined; the description and details of the environment in used in Appendix A.3;
- Another crucial concern is that the proposed reward scaling technique is not equivalent to the learned latent reward in the sense of resulting in the same optimal policies (it is not a policy invariant reward shaping [2]). A minimal example that illustrate this is the following MDP:
    - Imagine an MDP with 11 states (s_0 to s_10). Initial state s_0. There are two actions, a_1, a_2. If the policy takes a_1, it goes to s_1 and receives r_1 = 1000. If it takes a_2, it goes to s_2 and receives r_2 = 100. From s_3, regardless of the action taken, the agent goes to s_1 -> s_3 -> s_5 -> s_7 -> s_9 (terminal), and does not receive rewards in any of these states. Similarly, from s_2, it goes to s_2 -> s_4 -> s_6 -> s_8 -> s_10 (terminal), but receives reward +100 at each state. Assuming no discount in the return, we have V(s_1) = 1000 and V(s_2) = 500, thus the optimal policy should choose a_1. However, if you consider the reward scaling proposed in this work and assume a uniform distribution over states to compute the expectation in L211, then V(s_1) ~= 17/9 and V(s_2) ~= 30/9, which is a different optimal policy.
    - The reward shaping seems to work empirically at least for the Maze Navigation (Ravens-Manipulation is slightly worse and it is unclear in the LLM experiments if the VPL version leverages the scaling or not). It is important to state this caveat, since reward scaling implicitly assumes policy invariance under reward transformation.
- One minor concern is regarding the data requirements during test-time inference. During deployment, VPL would require interacting with a user to collect a few labels before performing inference. In the LLM experiment with GPT2, it looks like it requires 100 samples, so it is questionable if this is sample-efficient enough for interacting with humans in the real world. Is this experiment using the active inference from Section 4.2?


**Summary**: I believe the paper brings a contribution while presenting the failure of previous methods in pluralist alignment, and showing how they fail. Furthermore, the proposed methodology is interesting and seems to work well in small cases. Nevertheless, the presented empirical evidence does not support the scalability claims in LLMs, as the datasets are too simple and even raises questions whether very simple models could solve it. Furthermore, the reward scaling technique proposed is not well grounded theoretically and leads to different optimal policies. There are also some potential societal consequences that are not discussed. Therefore, I do not support acceptance in the current form of the paper, but. I am open to changing my score in case of the raised concerns being addressed or any misunderstanding clarified.

**Questions:**

- How does VPL handle irrational users? (i.e., users whose preferences are inconsistent across context/responses)
- Do the LLM experiments leverage the active inference from Section 4.2?
- Why isn’t VPL-SPO shown in Table 1?

**Limitations:**

- I believe the experimental setup does not support any claims of scalability with LLMs and it is unclear whether the proposed method would work in a scenario with multiple users and several variables affecting preferences, which is often the case in the preference datasets.
- There are some concerns in terms of social impact that are not discussed in Section A.5. A well-performant latent model results in a very personalized language policy, which can potentially lead to two main problems: first, the model being biased/presenting sycophancy [3] to satisfy user beliefs; second, the model reinforcing users with antisocial/unethical/criminal preferences, which often presents preferences that are very far from the rest of the society. Both scenarios are potentially harmful and would be interesting to discuss ways to mitigate them.


**References**:

[1] Siththaranjan et. al. Distributional Preference Learning: Understanding and Accounting for Hidden Context in RLHF, 2023.

[2] Ng et. al. Policy Invariance under Reward Transformations: Theory and Applications to Reward Shaping. ICML, 1999.

[3] Sharma et. al. Towards Understanding Sycophancy in Language Models, 2023.


# Post-Rebuttal

Please refer to my comment under the rebuttal message. Based on that, I am increasing my scores and recommending acceptance.

---

> ### Author Rebuttal · Authors · 2024-08-07
>
> Thank you for your feedback. We appreciate that you found the method interesting and agree that pluralistic alignment is a challenge that is mostly overlooked by the RLHF community.  We have conducted 5 additional experiments to demonstrate scalability and include the results in the rebuttal PDF, and address your concerns below.
>
>
> > **“The major concern is on the evaluation setup. [...] not sufficient to claim scalability in the LLM setup for the proposed method.”**
>
> In our LLM experiments, we adopt the widely used UltraFeedback dataset to expose pluralistic alignment. The original dataset has 4 attributes along which fine-grained preferences are generated, and in the original submission, we used two of these attributes to indicate diverse users (similar to prior work [2]). **We present additional experiments in Table AM:1 where we compare VPL and the baselines, in a scaled and more diverse setting where all four attributes are present in the preference dataset.** Effectively, in these experiments, **we go beyond the prior work** [2] that includes two users (harmlessness and helpfulness), and include all four attributes in the UltraFeedback dataset. This dataset presents a standard benchmark, which is similar to past works in preference modeling with diverse users [2, 3], as well as state-of-the-art preference modeling works that use either UF or HH-RLHF [4,5,6]. By using all available attributes in UF as distinct users, **we have created a challenging benchmark for pluralistic alignment from the best available real RLHF language datasets.**
>
> The reviewer suggested : “Although the paper brings a diverse [...] hidden context is composed of very few variables with simple distributions”. So, to demonstrate VPL in the presence of more users, we include new experiments with a large number of users and hidden variables, in Figure AM:2. We supplement the Maze and Habitat experiments with 10 and 5 users (Figures 8 and, 4 respectively), **with new Habitat-Rearrange experiments, where we have ~ 100 users**. Each user has a ranking over five different locations for storing objects in their home. **Here, the space of users and variables is combinatorial (as total possible orderings are 5!)**. In Figure AM:4, we observe that VPL infers the user distribution and adapts the downstream model, **demonstrating its effectiveness with a large user population**. Additionally, in Figure AM:3, we test the generalization of VPL to unseen users at test time and show that it can effectively infer and follow unseen goals at test time, outperforming the baselines.
>
> > **“Another crucial concern is that the proposed reward scaling technique is not equivalent to the learned latent reward [...]”**
>
> We appreciate the reviewer for taking the time to craft this example. To test the given MDP under the scaling from SPO [1], and standard version (BTL) of preference learning [13], we simulated the MDP you suggested and report the value function of states $s_1, s_2$.
>
> |                   | $V(s_1)$ | $V(s_2)$ |
> |-------------------|----------|----------|
> | Ground Truth      | 1000     | 500      |
> | Markovian BTL     | -18.87   | 22.70    |
> | Markovian SPO     | 17/9     | 30/9     |
> | Non-Markovian BTL | 36.46    | -33.36   |
> | Non-Markovian SPO | 35/9     | 10/9     |
>
> **Under a Markovian oracle, both methods fail to generate the correct value function i.e. $V(s_1) < V(s_2)$.** This points to a larger issue in preference learning, which all RLHF methods face and is not unique to our scaling. Essentially, in preference-based learning, we only have binary labels, and no notion of reward scale; so we assume how the scale of rewards affects the preferences (e.g. BTL or SPO). Consequently, under the assumed scale, a cumulative reward estimate (i.e. the value function) might not have the same ordering as the ground truth, which is what we observe in the above table.
>
> We adopt a common assumption in preference learning, particularly in [1] that inspired the scaling: the oracle providing preference labels is non-Markovian (i.e. it prefers all states in a trajectory with higher returns over those in a lower return one). **As a result, in the given MDP, the states $s_1, s_3, s_5, s_7, s_9$ would be preferred over the states $s_2, s_4, s_6, s_8, s_{10}$. This generates the accurate value functions and policy i.e. $V(s_1) > V(s_2)$, and the policy under this scaling is invariant.**
>
> > **“data requirements during test-time inference [...] it looks like it requires 100 samples, so it is questionable if this is sample-efficient enough for interacting with humans in the real world”**
>
> We would like to clarify the training and evaluation setup here:  **VPL uses only a random sample of 2-8 questions from the subset of 100 questions in the LM experiments to predict the latent distribution**, which provides us with a sample efficient way to infer the user distribution at test-time, as opposed to using all 100 samples.
>
>
> > **“How does VPL handle irrational users? (i.e., users whose preferences are inconsistent across context/responses)”**
>
> In Figure AM:2 of the rebuttal PDF, we include experiments with varying number of labels queried from each user at test time. We also add noise to the context i.e. we flip the labels with a certain probability to simulate inconsistent and noisy users. **The performance degrades with increasing noise, but VPL still outperforms the baseline at 25% noise level.  With more labels, VPL performs better at higher noise levels,** but at 50% noise, the context provides equal information for all user types. Consequently, it fails to identify the particular user and the performance coincides with the baseline (that has no mechanism for personalization). This shows that VPL is able to handle irrational users i.e. users whose preferences are inconsistent across context/responses.

---

> ### Author Response · Authors · 2024-08-07
> **Continued Rebuttal**
>
> > **the model being biased/presenting sycophancy [12] to satisfy user beliefs; second, the model reinforcing users with antisocial/unethical/criminal preferences, which often presents preferences that are very far from the rest of the society.**
>
> Thank you for bringing up this insight, and we will include the following discussion in the social impact statement.
>
> In pluralistic alignment, we assume that some differences in user preferences reflect divergent but equally valid perspectives on which moral or cultural values an LLM should align to; for example, individuals from one culture may hold collectivist moral values, while another culture is more individualist. Both value systems should be respected, and as such LLMs should be able to recognize and adapt to the values held by a particular user. However, the personalized model could potentially either be sycophantic or align with adversarial users, which is undesirable. This raises very interesting questions, such as: At what point should the LLM embrace a more universal set of values? How can we detect when such a point has occurred in a particular conversation? The probabilistic framework of the user distribution could allow us to identify low probability or rare behavior, and also the distributional nature of reward functions can help us point out responses where the users are divergent (maybe signifying disagreement). Additionally, a model could flexibly switch between adhering to the user's personal preferences and conforming to a more universal perspective on topics where it could be biased, or is sensitive to jailbreak [14]. Taking inspiration from Constitutional AI [14], we can allow a system designer to specify the topics for which the LLM should not engage in user personalization. Overall, this presents an exciting future research direction toward building safe personalized LLMs.
>
> > **Pets dataset is simple**
>
> We acknowledge that **Pets dataset is a simple synthetic dataset**. Therefore, we present the improved capabilities of our method with additional experiments on a larger and more diverse UltraFeedback dataset (UF-P-4),  with four different users (In Table AM:1). The Pets dataset provides a **sanity check over VPL** to show that the model is able to adapt to multi-modal preferences in imbalanced language datasets.
>
> > **“this raises concerns if variational inference is required here - perhaps a base LLM can already separate these users in the feature space, but there is no such a baseline in the work.”**
>
>   We include an additional baseline in Table AM:1, "VPL+Ground Truth User", where we replace the predicted latent user vector by the ground truth one-hot user vector to adapt the model to different users at training and test-time. **We show that VPL provides comparable performance to this condition (63% vs 66%)**. Here, a key advantage of VPL is that **it does not assume access to explicit user types** but learns to encode and cluster users directly from preference labels, while achieving similar accuracy.
>
> > **there is only preference modeling and no RLHF/alignment**
>
> We present experiments on learning policies using VPL reward modeling in simulated control domains. We acknowledge that our language experiments are focused on preference modeling, but we primarily follow and compare to prior work, which also focuses on preference modeling from diverse datasets alone [2, 3], without including experiments for LLM fine-tuning. We believe that preference modeling alone is interesting, as modeling diverse users may also provide increased interpretability in highlighting potential reasons for preferences [7, Sec. 2]. We believe that further exploring how to best apply VPL to downstream tasks and larger, noisier datasets is an interesting and exciting avenue for future research. Additionally, prior work has shown that improving RM performance can yield improved downstream performance, both when used in RLHF training [8, 9, 11] and when used in best-of-N settings [9, 10].
>
> > **“the requirement of filtering out the context where the users agree..”**
>
> In the additional experiments with four users (Table AM: 1 of the rebuttal PDF), we filter out instances only where all users disagree. So, the context can still contain queries where at least two users overlap. Thus, **VPL works in cases where different users agree on some responses, but not all of them.**
>
> We will add the following to the limitations sections of the paper: “In LLM preference modeling, VPL assumes that the queries used to generate context inputs for posterior inference contain some useful information to identify the users. In our work, we filter out the instances where all users agree i.e. avoiding degenerate contexts that provide no information about the users.”

---

> ### Author Response · Authors · 2024-08-07
> **Continued Rebuttal**
>
> > **"Methodology details"**
>
> Thank you for pointing out the missing details. We will update them clearly in the manuscript, as follows:
>
> 1. We assume that our prior is a multi-variate Gaussian with mean $\mu$ and covariance $\Sigma=\text{diag}(\sigma\sigma^T)$, where $\mu, \sigma \in \mathrm{R}^d$. In all experiments, they are initialized from a standard Gaussian. However, in our control experiments, we observed that using a learned Gaussian i.e. setting $\mu$ and $\sigma$ to learnable parameters under the ELBO objective (Eq. 3) improved performance and stability during training.
> 2. The humans are simulated using Oracle reward functions that are included in Appendix B.1. We randomly sample a user type and query a batch of annotations from the given user to generate a single data point.
> 3. The accuracy of the reward model is "1" if the LM rewards model assigns a higher reward to the response preferred by the given user over the alternative response, and "0" otherwise. We report the average accuracy of the model over the eval set consisting of multiple prompt and response pairs, labeled by diverse users.
> 4. VPL-SPO is required as a part of policy optimization while currently, we focus on preference modeling for LLMs in this work. VPL and VPL-SPO will model the preferences with similar accuracy. The only difference would be the scale of rewards that affects policy optimization.
>
>
> --
>
> [1] Swamy et al. (2024). A Minimaximalist Approach to Reinforcement Learning from Human Feedback.
>
> [2] Siththaranjan et al. (2023). Distributional Preference Learning: Understanding and Accounting for Hidden Context in RLHF
>
> [3] Zhao et al. (2023). Group Preference Optimization: Few-Shot Alignment of Large Language Models.
>
> [4] Ivison et al. (2023). Camels in a Changing Climate: Enhancing LM Adaptation with Tulu 2.
>
> [5] Rafailov et al. (2023). Direct Preference Optimization: Your Language Model is Secretly a Reward Model.
>
> [6] Cui et al. (2023). UltraFeedback: Boosting Language Models with Scaled AI Feedback.
>
> [7] Sorensen et al. (2024). A Roadmap to Pluralistic Alignment.
>
> [8] Shen et al. (2023). The Trickle-down Impact of Reward (In-)consistency on RLHF.
>
> [9] Gao et al. (2022). Scaling Laws for Reward Model Overoptimization.
>
> [10] Ivison et al. (2024). Unpacking DPO and PPO: Disentangling Best Practices for Learning from Preference Feedback.
>
> [11] Meng et al. (2024). SimPO: Simple Preference Optimization with a Reference-Free Reward.
>
> [12] Sharma et al. (2023). Towards Understanding Sycophancy in Language Models.
>
> [13] Ouyang et al. (2022). Training language models to follow instructions with human feedback.
>
> [14] Bai et al. (2022) Constitutional AI: Harmlessness from AI Feedback.

---

> > ### Comment · Reviewer_hKQW · 2024-08-12
> >
> > I appreciate the authors’ efforts on addressing questions and bringing new empirical evidence for the work.
> >
> > My major concern (about the empirical evidence not supporting the scalability claims) was properly addressed with the rebuttal experiments. There are now new experiments showing the method works for >100 users in Habitat Environment and an extension in the Ultrafeedback benchmark. The missing methodology details were also clarified in the rebuttal. I strongly recommend the authors to add these experiments and new clarification text for the camera-ready version, as it considerably improves the paper.
> >
> > My third concern was also clarified. In terms of the second concern (reward scaling), the rebuttal acknowledges the limitation and links it to an underlying assumption on the oracle’s preference labeling process (non-Markovian). I believe this assumption is strong and somewhat simplistic (if the ground-truth labeler only looks at the trajectory’s return and not for particular states, then the problem modeling could be simplified to a bandit setting instead of the full sequential decision-making setup - as it is often the case of traditional LLM-based RLHF). I believe the camera-ready version should also explicit this limitation/underlying assumption as a clarification.
> >
> > Overall, most of my crucial concerns were addressed and the paper largely benefited from the new content. Therefore, I am raising my score accordingly.
> >
> > Other points (questions/limitations) were properly discussed as well. Again, I strongly recommend authors to incorporate this on the camera-ready version, particularly the discussion regarding the societal impact.
> >
> >
> > (For the next time, **please make sure to adhere to the rebuttal length restriction**).

---

### Official Review · Reviewer_Z9tQ · 2024-07-13

**Soundness:** 3
**Presentation:** 2
**Contribution:** 3
**Rating:** 4
**Confidence:** 4

**Summary:**

The primary objective of the paper is to design a multi-modal RLHF strategy to align diverse preferences with a latent variable model. The latent variable represents the users/topics and the reward model conditioned on the latent variable is learned for each user preference. The empirical results support the hypothesis on simulated control problems and pluralistic language datasets.

**Strengths:**

1. Aligning to diverse preference with variational inference is one of the most natural ideas and the work provides an interetsing step in that direction.
2. The approach also provides a method to actively learn user preferences leveraging the posterior uncertainty.
3. The empirical performance and ablation shows that learning under the probabilistic framework is able to precise multimodal reward model.

**Weaknesses:**

1. The setup is not extremely practical. For ex: If I am the LLM company when a new user comes to the system and asks a question via the prompt? Then, instead of answering this, we will be doing some active learning to efficiently identify the user right? Since the posterior q(z|y_1, y_2) needs to be estimated for that? Is there any other way to do that or we need to incorporate the active learning strategy every time for each new user/group?
2. The experimental setup is restricted to a simulated environment and tasks. It is crucial to understand the efficacy of the approach when scaled to more realistic environments and tasks [1, 2, 3]
3. The work lacks comparison with several baselines on multi-objective RL or aligning with diverse preferences. Please provide a detailed comparison with them, not necessarily through experiments but at least a detailed discussion will be crucial. [1, 2, 3, 4, 5]


References :
1. MaxMin-RLHF: Towards an equitable alignment of large language models with diverse human preferences

2. RLHF from Heterogeneous Feedback via Personalization and Preference Aggregation

3. Modular Pluralism: Pluralistic Alignment via Multi-LLM Collaboration

4. Pareto-Optimal Learning from Preferences with Hidden Context

5. PERSONALIZED SOUPS: PERSONALIZED LARGE LANGUAGE MODEL ALIGNMENT VIA POST-HOC PARAMETER MERGING

**Questions:**

1. Can the authors motivate a practical scenario of the setup, where this setup can make sense?
2. It's not very clear on the specific nature of the prior and posterior being used for each of the tasks.
3. Also it is done on majorly simulated tasks and environments. Hence, it is important to show the performance in more realistic environments and benchmarks.

**Limitations:**

Check above

---

> ### Author Rebuttal · Authors · 2024-08-07
>
> We thank the reviewer for providing useful feedback regarding our work. We appreciate that the reviewer believes that our introduced approach is natural towards aligning models to diverse preferences. The reviewer brings up useful concerns regarding the scale of the experiments, demonstrating the effectiveness of VPL in realistic benchmarks and drawing bridges to practical scenarios for this problem and solution. We address these questions by providing additional experiments, and outline those and other responses below.
>
> > **“The experimental setup is restricted to a simulated environment and tasks.”**
>
> Thank you for suggesting that we scale our experiments to more realistic tasks and environments. Following this, we include five additional experiments in the rebuttal PDF:
> 1. In Table AM:1, **we scale the language experiments to UltraFeedback dataset with four users,** which is comparable in scale and diversity to state-of-the-art prior work on preference modeling [6]. In these new experiments, we go beyond the prior work [6] that includes two users (harmlessness and helpfulness), and include all four attributes in the UltraFeedback dataset. By using all available attributes in UF as distinct users, we have created a challenging benchmark for pluralistic alignment from the best available RLHF language datasets.
> 2. In Figure AM: 1, **we expand our robotics experiments to additional tasks in the Habitat simulator.**  We include a new Habitat-Tidy task inspired by TidyBot [7], that presents **a realistic setting for diverse users in the real world.**
> 3. To test VPL with a large number of users, in Figure AM: 4, we demonstrate that it outperforms the baselines in the Habitat-Rearrange task (when scaled to ~ 100 users).
> 4. We also show that VPL is able to generalize to new users at test-time i.e. adapt the model to align with unseen goals in Figure AM: 3.
> 5. In Figure AM:2, we show that this learned model is robust to noise or inconsistency in user context labels when modeling diverse user preferences.
>
> Overall, the additional experiments and analysis, along with the existing results cover a diverse number of realistic tasks with increasing scale and diversity.
>
> > **Can the authors motivate a practical scenario of the setup, where this setup can make sense?**
>
> Consider real-world tasks such as autonomous household robotics. **Each person has strong individual preferences for things like how a user arranges things in their home [7], which must be modeled by a robot to effectively assist in cleaning the home (such as one person may prefer storing shirts in the drawer, while another may prefer them on the shelf). In assistive robotics for the disabled** [10], different users have different physical constraints and preferences regarding feeding methods and food choices. To succeed at these tasks, the robot must efficiently infer and adapt to each user's unique preferences. Our simulated control experiments, as well as the additional experiments shown in Figure AM:1 in the rebuttal PDF, test whether VPL is a promising method for this type of real-world robotics task.
>
> LLMs must interact with a large, diverse, global population of users. Within this population, individuals from one culture may hold collectivist moral values, while another culture is more individualist [8]. Or in certain cultures, it is morally acceptable to eat a certain type of fish while pregnant, but in some others, it is a taboo. **Both of these value systems should be respected, and as such LLMs should be able to recognize and adapt to the values held by a particular user.**
>
> In our experiments, the UltraFeedback dataset contains multiple attributes that users may prefer, such as harmlessness and helpfulness [9]. E.g. if the prompt is “teach me how to dive into a river” - users can prefer the models to be either helpful (i.e. provide the steps to learn diving) or harmless (i.e. warn the user that it is dangerous and provide no instructions). Here, the LLMs should infer and personalize to the values and preferences of each user, and VPL presents a step in that direction.
>
> In summary, these directions present some practical cases where an AI model has to personalize and cater to multiple users. And, **our method presents a practical approach to this problem where each user answers a few questions to personalize the downstream model.**
>
> > **“The setup is not extremely practical. For ex: If I am the LLM company when a new user [...] any other way to do that or we need to incorporate the active learning strategy every time for each new user/group?”**
>
> We thank the reviewer for bringing up this concern. We would like to present an important clarification that the **active learning procedure needs to be performed only once after the model has been trained** to generate the most discriminative queries i.e. using active learning we get a set of N query pairs. Beyond active learning, we can **instead select random queries** (potentially leading to inefficient performance). For each new user, we get labels over the N pairs, which enables us to estimate the posterior $q(z | (s^i_1, s^i_2, y^i)_{i=1}^N)$. **Our experiments show we get accurate performance in as little as 2 test-time queries** (see Figure AM:2 in the rebuttal PDF and Figure 5 in the original submission). However, if we do not have access to any test-time queries for a new user, we have access to the prior $p(z) \sim \mathcal{N}(\mu, \sigma^T \mathrm{I})$, and **can sample from the mode of the prior,** where the model would respond according to the majority group (as in existing preference learning methods). Thus with no additional information, VPL is as performant as a vanilla RLHF model.

---

> > ### Comment · Reviewer_Z9tQ · 2024-08-12
> > **Response to Rebuttal by Authors**
> >
> > Thanks for providing clarifications to my concerns. I agree with the motivation for robotics, although for the LLMs still its not very clear. So, to be clear once you have the N pairs, with which you perform the active learning. But, what does it mean that the active learning is done once is not clear? Can you explain this in more detail.
> > Also, can you please specify the prior and posterior used for the robotics and LLM case?

---

> > > ### Author Response · Authors · 2024-08-12
> > >
> > > Thank you for engaging in discussion with us. We provide additional clarification regarding the motivation and practicality for VPL with LLMs. We also include a detailed explanation of the active learning approach and the prior/posterior structure (in addition to the outline in Section 4).
> > >
> > >
> > > ### ***LLM Motivation***
> > >
> > > As we show in our new experiment for reviewer PuPi, by default our method gives the same reward modeling performance as the current standard BTL model if there is a single user in the dataset or if we have no additional information from the user. So, if we have no additional preference labels from a user, our technique will not hurt performance. **However, if we can obtain 2-8 annotations from a user about which response they prefer, we can personalize our reward model to their specific preferences and values, unlocking the benefits of pluralistic alignment of LLMs.**
> > >
> > > ### ***Active Learning***
> > > We would like to clarify the active learning workflow in detail:
> > > 1. We have a set of queries  $(s^i_1, s^i_2)^K_{i=1}$, which is a pair of states (or responses to a given prompt, in case of LLMs). To create the training set, we sample a batch of queries of size $N$, and randomly ask one of the users to annotate it with their preferences ie. we get $\mathbf{S_j} = [(s^i_1, s^i_2, y^i)_{i=1}^N]_j$, where $N << K$.
> > > 2. We get multiple annotated batches from the diverse users to form the training set for the reward model i.e $D = (\mathbf{S_1}, \mathbf{S_2} \dots)$.
> > > 3. We train the reward model as indicated in Algorithm 1, and obtain the encoder $q(z | \mathbf{S_j})$  (i.e it takes as input a subset of queries $\mathbf{S_j}$) and the reward model R(s,z). The encoder output is $q(z | \mathbf{S_j})$, which is multi-variate gaussian distribution approximating the distribution over user preferences / groups or types.
> > > 4. ***Here, we start the active learning process***. Given $q$ and all the queries $(s^i_1, s^i_2)_{i=1}^K$, we generate multiple subsets of size N (total possible samples are $^N C_K$). For each given subset, we can find the information gain over the posterior in Step 3 and Eq. 4.
> > > 5. We choose the subset of N pairs $\textbf{S}_{active} = (s^i_1, s^i_2)_{i=1}^N$ with the max information gain. This is the set of questions that are most informative about the user type or distribution.
> > > 6. Finally,  **at evaluation for all incoming users, we ask them to annotate the same set of questions in $\textbf{S}_{active}$, and then provide it as input to the encoder to obtain the posterior over this user's preferences**.
> > >
> > > **Therefore, the entire process of active learning has to be done only once after training to obtain $\mathbf{S}_{active}$. At eval time, we just need a users to provide labels for the same $N$ pairs to predict the posterior distribution for reward / policy conditioning.**
> > >
> > >
> > > ### ***Posterior and Prior:***
> > > 1. We use a gaussian prior and an MLP based posterior similar to the approach in a standard VAE [1].
> > > 2. **The prior is a standard multi-variate gaussian of dimension** $d$, where the mean $\mu=[0 \dots 0]^T$ and the covariance $diag(\sigma \sigma^T), \text{ where } \sigma=[1 \dots 1]^T$. Here, $\mu, \sigma \in \mathrm{R}^d$; $d$ is the size of the latent dimension.  In the robotics experiments, we set them to be learnable parameters by setting the requires_grad property = True in pytorch.
> > > 3. For, **the posterior in robotics and LLMs we predict two vectors** $\hat{\mu}, \hat{\sigma} \in \mathrm{R}^d$  **using the encoder q and the annotated pairs $\textbf{S}_j$ as input**. It generates the posterior, which is also a multi-variate gaussian with the mean $\hat{\mu}$, and the covariance $diag(\hat{\sigma} \hat{\sigma}^T)$
> > > 4. The encoder architecture in the robotics case is a simple MLP that takes in the annotated pairs $\mathbf{S_j} = [(s^i_1, s^i_2, y^i)_{i=1}^N]_j$ i.e. the input is of dimension (2S+1)*N, where S is state dimension, and N is the number of queries.
> > > 5. For the LLM encoder architecture, we lay it out in detail in Section 5 and Figure 2 of the paper. It is similar to the robotics case, with certain modifications to handle the high dimensional and complex LLM embeddings.
> > > 6. For robotics, we sweep over possible values $d$ in { $8,16,32$ }, and for LLMs, $d=512$. We include the detailed hyperparameters in Appendix B.4
> > > 7. Finally, to predict the reward or condition the policy on the user type/preference, we sample a vector $z$ from the predicted posterior, and augment the input to the reward model or the policy to generate personalised rewards / behavior.
> > >
> > > [1] Kingma et al. (2013). Auto-Encoding Variational Bayes.
> > >
> > > Thank you for the suggestions and we will make this workflow more clear in the paper.

---

> ### Author Response · Authors · 2024-08-07
> **Continued Rebuttal**
>
> > **“comparison with several baselines on multi-objective RL or aligning with diverse preferences. ”**
>
> We thank the reviewer for providing additional references for our work.  We will incorporate the additional references in the related works section as follows:
>
> [1, 4] aims at trading conflicting alignment among diverse users with different objectives through techniques like Pareto-optimal optimization or multi-objective RL. The goal of such methods is to optimize the reward model to maximize worst-case performance over the different groups. In contrast, **our work does not aim to optimize against the diversity but rather solve the model misspecification** and learn reward models that can infer the context and specialize to a particular user. This ensures the model can align to all user groups, rather than trade-off among them.
>
> [2] introduces an approach that uses explicit clustering of human groups and learns individual reward functions for the different clusters. Our work instead relies on variational inference and latent conditioned rewards to infer and model diverse humans directly from the preference data.  [2] further introduces an additional method that assumes a single reward function for all the clusters, but **adopts a probabilistic approximation to the reward model, similar to DPL [6]. We include DPL as a baseline in all the LLM experiments** in Table 1 in the original submission and Table AM:1 in the rebuttal PDF, and show that our method outperforms DPL across multiple datasets.
>
> We included a reference to [5] in L112 of the original submission.
>
> [3] proposes a pluralistic alignment framework, **using smaller LMs that are trained on community or user-specific data**. Further, it uses responses from the smaller community LMs to adapt a larger LLM to provide responses covering all or just one specific user. Meanwhile, **our approach adopts an unsupervised method (no access to explicit user distributions)** to identify the latent and condition the preference model towards the specific user preferences.
>
>
> > **prior and posterior structure**
>
> Here, we provide additional details about the structure of the prior and posterior of our model. We assume that our prior is a multi-variate Gaussian with mean $\mu$ and covariance $\Sigma=\text{diag}(\sigma\sigma^T)$, where $\mu, \sigma \in \mathrm{R}^d$. In all experiments, they are initialized from a standard Gaussian. However, in our control experiments, we observed that using a learned Gaussian i.e. setting $\mu$ and $\sigma$ to learnable parameters under the ELBO objective improved performance and stability during training. The posterior is an MLP that takes in the annotated samples $\textbf{S} \sim (s^i_1, s^i_2, y^i)_{i=1}^N$ and predicts the latent distribution $\mathcal{N}(f(\textbf{S}), g(\textbf{S}))$.
>
> > **Also it is done on majorly simulated tasks [...]**
>
> Our work presents **an algorithmic solution to the problem of personalization** in RLHF. We present extensive experiments across simulated control and language experiments, which we believe present realistic setups and benchmarks. So, we believe real-robot experiments to be beyond the scope of this algorithmic paper and leave on future work the challenge of deploying this method to real-world robot systems.
>
>
>
> --
>
> [1] Chakraborty et al. (2024). MaxMin-RLHF: Towards an equitable alignment of large language models with diverse human preferences.
>
> [2] Park et al. (2024). RLHF from Heterogeneous Feedback via Personalization and Preference Aggregation.
>
> [3] Feng et al. (2024). Modular Pluralism: Pluralistic Alignment via Multi-LLM Collaboration.
>
> [4] Boldi et al. (2024). Pareto-Optimal Learning from Preferences with Hidden Context
>
> [5] Jang et al. (2024). Personalized Soups: Personalized Large Language Model Alignment via Post-hoc Parameter Merging.
>
> [6] Siththaranjan et al. (2023). Distributional Preference Learning: Understanding and Accounting for Hidden Context in RLHF
>
> [7] Wu et al. (2023). TidyBot: Personalized Robot Assistance with Large Language Models.
>
> [8] Graham et al. (2013). Moral Foundations Theory: The Pragmatic Validity of Moral Pluralism.
>
> [9] Cui et al. (2023). UltraFeedback: Boosting Language Models with Scaled AI Feedback.
>
> [10] Bhattacharjee et al. (2020). Is More Autonomy Always Better? Exploring Preferences of Users with Mobility Impairments in Robot-assisted Feeding.

---

### Official Review · Reviewer_KQbQ · 2024-07-13

**Soundness:** 3
**Presentation:** 3
**Contribution:** 3
**Rating:** 5
**Confidence:** 3

**Summary:**

Instead of learning a unimodal reward model as in standard RLHF, this work aims to learn a reward that covers a diverse range of preferences. It assumes that user preferences are not explicitly given, such as through verbal descriptions in the prompt/instruction. Instead, preferences are implicitly provided through rankings among candidate responses. Technically, it uses variational inference to learn an encoding that characterizes any user’s preferences. Accordingly, the policy model is conditioned on the learned preference latent code, making the generation steerable. The authors also discuss how to select the most representative set of response pairs for each user.

**Strengths:**

- The paper is well-written and easy to follow.
- The study is addressing a crucial problem. The setup is realistic, as users may not always want to explicitly state their preferences.

**Weaknesses:**

1. It is unclear how many labels are needed to accurately learn a user preference or profile encoder. There should have been experiments evaluating (a) how encoder’s performance improves as the # of user samples increases; and (b) how well the encoding generalizes — can it encode unseen user profiles with high fidelity, and can it extrapolate and interpolate?

2. The claim of being the first work that learns latent-code conditioned reward is not correct. [1] and [2] below also learn multi-modal reward.
    - [1] Guan, Lin, Karthik Valmeekam, and Subbarao Kambhampati. "Relative behavioral attributes: Filling the gap between symbolic goal specification and reward learning from human preferences." ICLR 2023
    - [2] Wu, Zeqiu, et al. "Fine-grained human feedback gives better rewards for language model training." NeurIPS 2023

3. The setup of the LLM experiment is quite simple -- the number of attributes or dimensions of preferences is very limited. While the feasibility of attribute-conditioned reward modeling has been demonstrated in previous works [1,2], this work doesn't significantly extend beyond them in terms of scalability.

4. One missing piece in the experiment is an analysis of the latent code's negative impact on the language model. An important feature of language models, especially LLMs, is their versatility. They are not supposed to only answer questions related to pets. One question that needs to be answered is whether conditioning on a pets-related latent code would lead to catastrophic forgetting or distortions in responses to other tasks/questions unrelated to pets.

5. I understand that Section 4.2 discusses the strategy for selecting the most representative state-pair sets during deployment. However, this process seems quite costly as it requires multiple full passes through the dataset. A complexity analysis should have been included.

6. Real-world user profiles and preference data are often unbalanced. The imbalance may affect the approach's effectiveness.

**Questions:**

See the Weakness section.

**Limitations:**

See the Weakness section.

---

> ### Author Rebuttal · Authors · 2024-08-07
>
> Thank you for the useful feedback. We appreciate that the reviewer recognizes the importance of the problem, and the realistic setup introduced to study it. The reviewer raised useful questions about the effectiveness of the user encoder with varying labels, the generalization performance, and the scale of experiments. Overall, the concerns raised are very interesting and we provide additional experiments to demonstrate the usefulness of our method in real-world settings. We respond to all the concerns individually below:
>
> > **[1] and [2] below also learn multi-modal reward**
>
> Thank you for pointing us to these references. While these methods present conditional reward models, we highlight the specific differences between our work and the references. We will update our related section to contrast our contribution with these works as follows:
>
> [1] presents an approach to **learning a reward model conditioned on a feature vector** that represents the preferences for the desired behavior. Further, in [1] the users provide feedback on the individual features of a trajectory. However, in our work, we do not assume access to such explicit features. Instead, we propose a method to encode and predict latent distributions directly from binary preferences in an unsupervised setting. **Moreover, [1] does not include experiments on language models and only validates the method on simulated locomotion experiments, while we scale our approach to state-of-the-art preference datasets for LLMs, along with diverse and realistic simulated experiments.**
>
> [2] introduces a method to use dense rewards over multiple specific attributes to align LLMs. **This treats the problem of pluralistic alignment as a multi-objective RL problem** i.e. learn different reward models for each user or objective individually. Whereas in our work we aim to encode and model the multi-modal user preferences, and then condition the same model i.e. steer it to align with the particular user.
>
> > **“this work doesn't significantly extend beyond them in terms of scalability.”**
>
> In addition to the differences highlighted above, we also scaled our experiments to demonstrate VPL. Following this, we include five additional experiments in the rebuttal PDF:
> 1. In Table AM:1, **we scale the language experiments to the UltraFeedback dataset with four users.** In these new experiments, we go beyond the prior work [5] that includes two users (harmlessness and helpfulness), and include all four attributes in the UltraFeedback dataset. By using all available attributes in UF as distinct users, we have created a challenging benchmark for pluralistic alignment from existing real RLHF language datasets.
> 2. Our method is more general and focuses on inferring the (potentially unknown number of) different preferences without explicit user information and learning a latent conditioned model to adapt to the particular end user. Meanwhile, [1] uses explicit features for modeling different users or goals, and [2] learns different reward models for the different features. **The probabilistic nature of VPL enables us to do active learning as well (Figure 5 in original submission), which makes this a more efficient model for diverse preferences.**
> 3. In Figure AM: 1, we expand our robotics experiments to additional tasks in the Habitat simulator.  We include a new Habitat-Tidy task inspired by TidyBot [7], that presents a realistic setting for diverse users in the real world.
> 4. To test VPL with a large number of users, in Figure AM: 4, we demonstrate **that it outperforms the baselines in the Habitat-Rearrange task (when scaled to ~ 100 users)**.
> 5. We also show that VPL can **generalize to new users** at test-time i.e. adapt the model to align with unseen goals in Figure AM: 3.
>
> Overall, the additional experiments and key differences highlight that our proposed approach presents a scalable and general method for pluralistic alignment over the included references. In summary,  [1,2] focuses on learning personalized policies or reward models given the explicit classes or users. Further, these methods are tested only over language or control experiments. In our work, **we present an initial step towards learning a probabilistic framework to predict the latent user distribution directly from binary comparisons for language models and learn personalized policies that are tested in dynamic and realistic control settings.**
>
>
> > **“how encoder’s performance improves as the # of user samples increases”**
>
> We present additional experiments, where we vary the number of labels at test-time, for LLMs (in Figure AM:2) and control experiments (in Figure 5 in the original submission). **It shows that the performance of the model improves with the number of labels, reaching peak performance at 8 labels**.  However, **actively selecting the queries could allow the model to perform similarly with fewer queries** (see Section 4.3 and Figure 5). Furthermore, using more labels makes the model robust to small noise levels as observed in Figure AM:2.
>
> > **“how well the encoding generalizes”**
>
> To test the generalization of VPL to new users, we include experiments in Figure AM:3, where the reward model and policy are trained on users preferring 10 different goals in the maze, while at test-time the agent interacts with **users preferring 5 unseen goals that are sampled in-distribution**. We compare the performance to an oracle baseline, that uses the exact goal information during training and test time, to analyze how well the encoder can infer the distribution over unseen users. **In Figure AM:3, we observe that VPL significantly outperforms the baselines, and has performance at par with the oracle baseline.** This demonstrates that VPL can interpolate between different users, even slightly outperforming the oracle that is trained on only the set of 10 goals, while VPL uses the prior to optimize the policy over the set of all possible in-distribution goals.

---

> ### Author Response · Authors · 2024-08-07
> **Continued Rebuttal**
>
> > **“This process seems quite costly as it requires multiple full passes through the dataset. A complexity analysis should have been included.”**
>
> Thank you for this comment, we would like to clarify **that our approach doesn't require multiple passes over the entire dataset for each new user**. In our active inference techniques, we use a sampling-based method inspired by [3] to actively query the model. Given a dataset of D queries $(s^i_A, s^i_B)_{i=1}^{|D|}$, we sample $S$ query batches of size $Q$, where $Q$ is number of annotated samples per batch we get from a user. Here, $Q \in [2,8]$, **so we need to perform O(S * Q)** passes over the model with batch size $2^Q \sim [4, 256]$. Furthermore, this process only needs to be performed once after the model is trained to obtain the most discriminative set of queries for the given model. Finally, whenever a new user interacts with the system, we need to get labels on the actively inferred queries (usually 2-4) but do not require any additional passes over the query dataset.
>
> > **“The imbalance may affect the approach's effectiveness.”**
>
> Thank you for your attention to this detail. In our control and language experiments, particularly the control experiments and the pets dataset are imbalanced i.e. the preference dataset contains an unequal number of preferences from individual groups or users. As a result, the baselines can achieve > 50% accuracy, converging to the preferences of the majority user groups. So, **VPL works in the presence of imbalanced datasets.**
> While recent works [7, 8] focus on achieving Pareto-optimal performance across the groups, VPL treats each user individually via latent conditioning and does not suffer from this problem. VPL is able to personalize to the minority groups as well (see Figure 11 in Appendix A).
>
> > **One missing piece in the experiment is an analysis of the latent code's negative impact on the language model. An important feature of language models, especially LLMs, is their versatility. They are not supposed to only answer questions related to pets. One question that needs to be answered is whether conditioning on a pets-related latent code would lead to catastrophic forgetting or distortions in responses to other tasks/questions unrelated to pets.**
>
>
> Thank you for raising this insightful question. VPL introduces a latent bottleneck during the reward learning process, but since the base architecture of the policy and the reward model does not change otherwise, we do not believe this should majorly affect the performance of the model. We will leave a thorough investigation of this question to future work, but will note this as a potential limitation in the limitations section.
>
>
> --
>
> [1] Guan et al. (2023). Relative behavioral attributes: Filling the gap between symbolic goal specification and reward learning from human preferences.
>
> [2] Wu et al. (2023). Fine-grained human feedback gives better rewards for language model training.
>
> [3] Sadigh et al. (2017). Active Preference-Based Learning of Reward Functions.
>
> [4] Peng et al. (2024). Pragmatic Feature Preferences: Learning Reward-Relevant Preferences from Human Input.
>
> [5] Siththaranjan et al. (2023). Distributional Preference Learning: Understanding and Accounting for Hidden Context in RLHF.
>
> [6] Ivison et al. (2023). Camels in a Changing Climate: Enhancing LM Adaptation with Tulu 2.
>
> [7] Boldi et al. (2024). Pareto-Optimal Learning from Preferences with Hidden Context
>
> [8] Chakraborty et al. (2024). MaxMin-RLHF: Towards an equitable alignment of large language models with diverse human preferences.

---

> > ### Comment · Reviewer_KQbQ · 2024-08-11
> >
> > I thank the authors for their detailed response.
> >
> > First, I would like to point out that the rebuttal exceeded the character limit by being posted as an Official Comment instead of as a Rebuttal. I hope the authors can better adhere to the conference policy and respect the time of reviewers.
> >
> > While I don't think the rebuttal adequately addresses my concerns, I still find that the upsides of this work outweigh the downsides. Therefore, I would like to maintain my current positive rating.

---

> > > ### Author Response · Authors · 2024-08-11
> > >
> > > We would like to thank the reviewer for their response. We apologize for the response length and will be considerate of that in the future.  Could the reviewer please point us to specific questions to expand upon? This would help us address their concerns more effectively.

---

### Official Review · Reviewer_PuPi · 2024-07-15

**Soundness:** 4
**Presentation:** 3
**Contribution:** 3
**Rating:** 7
**Confidence:** 3

**Summary:**

This paper introduces a new framework for preference learning which tailors to user-preferences. Human feedback with Variational Preference Learning (VPL) learns a latent reward / preference for each user at the test-time. They furthur show potential application of techniques from active learning and uncertainty estimation to the framework.

**Strengths:**

- Paper is generally well-written and clear
- The research problem is very well motivated -- personalization to preferences that go beyond a universal notion of a single preference function.
- Method itself is novel and well-formulated to match the problem.

**Weaknesses:**

- My main concern lies in evaluation outside of designed control environments. To truly test the quality of a Reward Model (RM), one needs to show the downstream policy performance benefiting from improvements in the RM. These results are not present currenlty in section 7. The issue of reward variance as discussed in Section 4.1, may require further design decisions coupled with the optimization algorithm (PPO, REINFORCE, RLOO, online contrastive losses like DPO, etc.). Previous work has studied the general issue of gradient variance in RLHF (which is directly related to reward variance through the REINFORCE estimator) which suggest that this may not be an issue [1].


[1] Ahmadian et. al. "Back to Basics: Revisiting REINFORCE Style Optimization for Learning from Human Feedback in LLMs"

**Questions:**

Suggestions:

- Having a single-modal dataset, such as summarization, would help further ground the work and increase the experimental depth. The expectation is that we shouldn't see many benefits when using VPL compared to traditional RLHF.

**Limitations:**

Limitations have been specified by the authors.

---

> ### Author Rebuttal · Authors · 2024-08-07
>
> We appreciate the reviewer’s feedback and are glad that they agree with the motivation of the problem and the novelty of the approach. We address the concerns raised as follows:
>
> > **“Having a single-modal dataset [...] would help further ground the work and increase the experimental depth.”**
>
> This is an interesting analysis and we thank the reviewer for suggesting this. We ran additional experiments on the UltraFeedback dataset, where we considered the preferences of a single user (preferring the model to be “honest” over all other attributes) to analyze the single-modal case as suggested. **The standard BTL model gives a 77.04% eval accuracy while our VPL model gives a 77.16% eval accuracy. Our model matches the baseline performance, indicating that there is no drop in performance when using VPL compared to traditional RLHF over an unimodal dataset.**
>
> > **“evaluation outside of designed control environments.”**
>
> We present experiments on learning policies using VPL reward modeling in simulated control domains, which we have expanded and made more realistic in the rebuttal PDF. We acknowledge that our language experiments are focused on preference modeling, but we primarily follow and compare to prior work, which also focuses on preference modeling from diverse datasets alone [1, 2], without including experiments for LLM fine-tuning. We believe that preference modeling alone is interesting, as modeling diverse users may also provide increased interpretability in highlighting potential reasons for preferences [3, Sec. 2]. We believe that further exploring how to best apply VPL to downstream tasks and larger, noisier datasets is an interesting and exciting avenue for future research. Additionally, Prior work has shown that improving RM performance can yield improved downstream performance, both when used in RLHF training [4, 5, 7] and when used in best-of-N settings [5, 6].
>
> --
>
>
>
> [1] Siththaranjan et al. (2023). Distributional Preference Learning: Understanding and Accounting for Hidden Context in RLHF.
>
> [2] Zhao et al. (2023). Group Preference Optimization: Few-Shot Alignment of Large Language Models.
>
> [3] Sorensen et al. (2024). A Roadmap to Pluralistic Alignment.
>
> [4] Shen et al. (2023). The Trickle-down Impact of Reward (In-)consistency on RLHF.
>
> [5] Gao et al. (2022). Scaling Laws for Reward Model Overoptimization.
>
> [6] Ivison et al. (2024). Unpacking DPO and PPO: Disentangling Best Practices for Learning from Preference Feedback.
>
> [7] Meng et al. (2024). SimPO: Simple Preference Optimization with a Reference-Free Reward.

---

### Author Rebuttal · Authors · 2024-08-07

[Disclaimer: All Figures and Tables referred to as **AM:X** are in the **A**dditional **M**aterial.]

We thank the reviewers for their careful reading and constructive feedback. We appreciate that all reviewers recognize the importance and relevance of the problem of pluralistic alignment in preference learning. All reviewers agree that our realistic setup and experiments show the failure of the baselines, and our proposed method is an interesting and novel approach to the described problem.

Reviewers (hkQw, Z9tQ, KQbQ) had concerns about the scale of experiments for LM preference modeling and evaluation of our method in diverse environments. To address these concerns, we have conducted 7 additional experiments; the results of 5 are included in the rebuttal PDF, and additional results requested by reviewers hKQW and PuPi are included in the response to those reviewers. Below, we provide further context on the 5 experiments in the rebuttal PDF.

> **Scaling language model results to state-of-the-art preference datasets. (Reviewers hKQW, Z9tQ, KQbQ)**

In Table AM:1, we present additional results while scaling VPL to a version of the UltraFeedback dataset, where we consider users preferring all four diverse attributes. Here, as in prior work [1], we treat the rating attributes (harmless, helpful, etc.) as distinct users. **Going beyond prior work([1] considers only two users), we scale to 4 users, and test whether we can steer the reward model to personalize to the particular user.** As reviewer hkQw mentioned, this further presents a **larger and more complicated dataset** (that is comparable in scale to prior works[1,2,3,4]) to infer the user type and predict the downstream rewards. At the time of writing, the standard and best available RLHF datasets used in prior work on preference modeling are HH-RLHF [5] and UF [6]. By using all available attributes in UF as distinct users, we have created a challenging and scaled benchmark for pluralistic alignment from existing real RLHF language datasets.

In Table AM:1, we observe that VPL outperforms all the baselines in terms of accuracy, leading to a 10% improvement. VPL performs comparably to an Oracle baseline with explicit user information. **This highlights the ability of VPL to scale to larger datasets with more users, and richer prompts and responses.**

> **Scaling to dynamic and realistic home environments. (Reviewers Z9tQ, KQbQ)**

To apply VPL to a set of complex control environments, and practical settings for robots interacting with diverse users, **we include additional experiments in the Meta Habitat Simulator[7].** Inspired by TidyBot [8], we include experiments in a realistic home environment, where the robot has to pick and place objects around the house, personalizing the cleanup based on user preferences (particularly, storing kitchen goods based on their attributes i.e. putting plastic items together and metal items together, or do it for tableware/kitchenware). VPL is able to infer the user’s preferences and steer the robot to the desired behavior with a 30% higher accuracy. Meanwhile, the baselines collapse to a single solution that doesn't cater to the diverse users, as shown in Figure AM:1. **We believe this additional experiment illustrates both the need for personalized preference learning for applications like autonomous household robotics and the fact that VPL is a promising method for achieving it at scale.**

> **Scaling to many (∼100) users. (Reviewers hKQW, KQbQ)**

Currently, the UltraFeedback dataset with four users provides us with one of the largest and most diverse datasets for aligning models, to the best of our knowledge. So, to test the effectiveness of our approach in the presence of a larger number of users (not bounded by datasets), in Figure AM:4, **we include simulated experiments on a new Habitat-Rearrange task, where we have ~ 100 users.** Each user has a ranking over five different locations preferred for storing objects in their home. Here, the space of users and variables is combinatorial (as the total possible orderings are 5!). **This presents a challenging benchmark with a much larger number of users and hidden variables, and in Figure AM:4, we show that VPL consistently outperforms the baselines by ~ 30%** in aligning to user-preferred locations for the rearrangement task.

> **Effect of noise and number of queries from users at test-time. (Reviewers hKQW, KQbQ)**

In Figure AM: 2, we evaluate VPL in the presence of irrational users i.e. how does VPL perform when the users are noisy or inconsistent across their preferences? To simulate this, we progressively add more noise to the context i.e. flip the preference labels in the context with increasing probability. We also test the effect of the number of labels provided by a user, on the downstream performance.

We make the following observations:
1. The performance of **the model degrades with increasing noise, however, VPL still significantly outperforms the baselines** with noisy and inconsistent preference labels from the users.
2. At 50% noise the context labels provide equal information to the encoder about all users. Consequently, it fails to identify the particular user and the performance coincides with the baseline (that has no mechanism for personalization).
3. **Eliciting more preference labels from users generates better latent estimates and improves the accuracy of the preference model.**
4. Additionally, it also makes the performance robust at high levels of noise. Overall, **VPL provides reasonable performance with as low as ~ 2 queries,** and we report the best performance at ~ 8 test-time queries.

Overall, these results show that VPL is able to "handle irrational users (i.e., users whose preferences are inconsistent across context/responses)”, as requested by reviewer hKQW. Further, this also shows that VPL works is test-time data efficient (i.e. shows reasonable performance with 2 user queries).

---

### Author Response · Authors · 2024-08-07
**Continued: Author Rebuttal**

> **Generalization to unseen users.(Reviewers KQbQ)**

In Figure AM:3, we compare VPL against the baselines in a setting where the agent is trained on preferences from users preferring goals from a set of 10 locations in the maze, but at evaluation, **it interacts with users preferring goals from a set of 5 unseen locations** (sampled in distribution). We include a ground truth baseline, conditioned on the exact training and test locations (so explicit goal information). We show that VPL can interpolate between users and pursue the unseen goals better than the baselines, performing comparably to the ones with ground truth information. **This provides evidence that VPL could adapt and personalize AI models to unseen users.**


> **VPL performance in unimodal settings (Reviewers PuPi)**

An interesting concern by reviewer PuPi was to show the performance of VPL in settings with an unimodal preference dataset. We ran additional experiments on the UltraFeedback dataset, where we considered the preferences of a single user (preferring the model to be “honest” over all other attributes) to analyze the single-modal case as suggested. The standard BTL model gives a 77.04% eval accuracy while our VPL model gives a 77.16% eval accuracy. Our model matches the baseline performance, indicating that **there is no drop in performance when using VPL compared to traditional RLHF over an unimodal dataset**.

> **Better explanation of the active learning approach to VPL (Reviewers hKQW, KQbQ)**

Here, we include additional details to clear a misunderstanding regarding the active learning experiments. In our active inference technique, we use a sampling-based method inspired by [9] to generate the active queries for the model. Given a dataset of D queries $(s^i_A, s^i_B)_{i=1}^{|D|}$, we sample $S$ query batches of size $Q$, where $Q$ is number of annotations per batch we get from a user. Here, $Q \in [2,8]$, so we need to perform O(S * Q) passes over the model with batch size $2^Q \sim [4, 256]$. Furthermore, this process only needs to be **performed once after the model is trained to obtain the most discriminative set of queries for the given model.** Finally, whenever a new user interacts with the system, we need to get labels on the actively inferred queries (usually 2-4) but do not require any additional passes over the query dataset. In our experiments (Figure 5), we show that using active learning allows the model to achieve comparable performance with fewer queries (\~2), as compared to randomly sampled larger (\~8) queries.


--

[1] Siththaranjan et al. (2023). Distributional Preference Learning: Understanding and Accounting for Hidden Context in RLHF

[2] Rafailov et al. (2023). Direct Preference Optimization: Your Language Model is Secretly a Reward Model.

[3] Zhao et al. (2023). Group Preference Optimization: Few-Shot Alignment of Large Language Models.

[4] Conitzer et al. (2024). Social Choice Should Guide AI Alignment in Dealing with Diverse Human Feedback.

[5] Bai et al. (2022). Training a Helpful and Harmless Assistant with Reinforcement Learning from Human Feedback.

[6] Cui et al. (2023). UltraFeedback: Boosting Language Models with Scaled AI Feedback.

[7] Puig et al. (2023). Habitat 3.0: A Co-Habitat for Humans, Avatars and Robots.

[8] Wu et al. (2023). TidyBot: Personalized Robot Assistance with Large Language Models.

[9] Sadigh et al. (2017). Active Preference-Based Learning of Reward Functions.

---

### Decision · Program_Chairs · 2024-09-25

**Decision:**

Accept (spotlight)

**Comment:**

This paper proposes a new approach to pluralistic alignment, a very important and topical problem. I agree with Reviewers hKQW and Z9tQ that using latent variable modeling is a natural approach; I'd even say it feels like the "right" approach. Unfortunately, there wasn't much substantiave back-and-forth during the discussion period, but the conversation with Reviewer hKQW seemed fruitful, and I would also encourage the authors to incorporate their responses into the camera-ready. Reviewer KQbQ's concerns do actually appear to be well-addressed, and Reviewer Z9tQ's remaining concerns don't seem like deal breakers. The proposed active learning approach is clearly effective, and I think it's both clever and well-explained. The fact that the experiments cover both simulated control tasks and LLMs is already impressive and shows the generality of the approach. Overall, this is an excellent paper that I expect to be of great interest to many NeurIPS attendees.